# Towards Pareto-Optimal Tool-Integrated Agents with Pareto Ranking Policy Optimization

**Junyi Li** [1] **Xiaowei Qian** [1] **Yingyi Zhang** [1] **Wenlin Zhang** [1] **Guojing Li** [1] **Sheng Zhang** [1] **Xiao Han** [2] **Yichao Wang** [3] **Xiangyu Zhao** [1]

## Abstract

Recent advances in tool-integrated language agents have significantly improved their ability to solve complex reasoning tasks. However, existing alignment methods predominantly focus on maximizing task accuracy, while overlooking auxiliary objectives such as tool-use efficiency, which are essential for practical deployment. To address this gap, we introduce **ParetoPO**, a two-stage multi-objective optimization framework for aligning tool-using large language models (LLMs) under competing objectives. In the first stage, ParetoPO leverages hypervolume-guided dynamic scalarization to adapt reward weights based on global Pareto frontier progress. In the second stage, it replaces scalarized learning signals with Pareto-ranking-based advantage computation, promoting nondominated trajectories through dominance-aware credit assignment. This design enables fine-grained, action-level optimization across multiple conflicting objectives. Experimental results on mathematic reasoning and multi-hop QA tasks show that ParetoPO consistently discovers policies with superior accuracy-efficiency trade-offs compared to static and heuristic baselines. Our code is publicly available at https://github.com/Applied-Machine-Learning-Lab/ICML2026_ParetoPO.

## 1. Introduction

Online reinforcement learning (RL) has become the de facto approach for aligning tool-augmented large language model (LLM) agents in complex tasks, from question answering with search engines to code generation with compilers (Zhao et al., 2023; Wang et al., 2024b). This paradigm has led to notable gains in task performance, but current alignment strategies predominantly optimize final-answer accuracy while neglecting auxiliary objectives at the process level (Li et al., 2025d; Jin et al., 2025; Zhang et al., 2026). In practice, factors like inference efficiency (e.g., number of tool calls) and step-wise decision quality directly impact resource usage and reliability, yet these are often omitted from the optimization process. In this paper, we mainly focus on *multi-objective RL* for LLM agents to not only produce correct solutions but also employ tools with high efficiency.

Existing approaches to multi-objective alignment in LLMs can be grouped into two main categories. (1) Fixed-weight scalarization or heuristic reward mixing: Many studies combine objectives via a weighted sum or ad-hoc reward interpolation using static weights (Wang et al., 2025a). However, such fixed-weight method is fundamentally limited. Different objectives often differ in scale and learning dynamics, i.e., a static weight that is appropriate early in training may later misallocate learning effort. Moreover, static linear scalarization can only recover Pareto-optimal solutions on the convex portions of the trade-off curve, missing solutions in non-convex regions (Lu et al., 2025). These issues leave fixed-weight methods prone to suboptimal results. (2) Multi-objective RL with gradient-based techniques: Other works pursue more sophisticated multi-objective optimization by computing separate gradients per objective and combining them (Li et al., 2025a; He & Maghsudi, 2025). While such methods formally accommodate multiple goals, they are often computationally expensive and have mostly been applied to simple settings. Importantly, most prior efforts focus on high-level semantic objectives (e.g., helpfulness, harmlessness) rather than action-level decisions of agents. Thus, balancing multiple behavioral objectives (like tool-use efficiency) alongside accuracy remains an open challenge.

In this work, we propose **ParetoPO**, a novel multi-objective

[1]Department of Data Science, City University of Hong Kong, Hong Kong, China [2]Zhejiang University of Technology, Hangzhou, China [3]Huawei Technologies Noah's Ark Lab, Hong Kong, China. Correspondence to: Xiangyu Zhao <xianzhao@cityu.edu.hk>, Yichao Wang <wangyichao5@huawei.com>.

*Proceedings of the 43rd International Conference on Machine Learning*, Seoul, South Korea. PMLR 306, 2026. Copyright 2026 by the author(s).

online RL method for LLM-based tool-using agents that addresses the above limitations. The proposed training framework dynamically adjusts the emphasis on each objective during learning and biases policy updates toward Pareto-optimal behaviors. Specifically, the optimization consists of two stages: (1) **Hypervolume-guided dynamic scalarization**, which adaptively tunes the reward weight of each objective in real time based on smoothed hypervolume signals, and (2) **Pareto-ranking-based policy optimization**, which uses non-dominated sorting for ranking a group of trajectories to estimate their advantage values, ensuring that improvements in one objective are not achieved at disproportionate expense of the other. From a global optimization perspective, the first stage facilitates efficient exploration of the objective space and guides the policy toward supported Pareto regions. Building on this coverage, the second stage refines policy updates by directly optimizing with respect to Pareto dominance, driving the policy toward Pareto-ascent stationary points where no common first-order improvement direction exists across objectives.

We conduct extensive experiments to evaluate the effectiveness of ParetoPO on both mathematical reasoning and multi-hop question answering tasks. The results show that our method achieves a superior trade-off between task performance and tool-use efficiency compared to existing single- and multi-objective baselines.

## 2. Background

### 2.1. Multi-Objective Markov Decision Process

We formulate our agentic tasks in the framework of a *Multi-Objective Markov Decision Process* (MOMDP), which generalizes the standard MDP to multiple reward signals. A MOMDP is defined as a tuple $\langle \mathcal{S}, \mathcal{A}, \mathcal{P}, \mathcal{R}, \boldsymbol{\gamma}, \mathcal{D} \rangle$, where:

- $\mathcal{S}$ is the state space, and $\mathcal{A}$ is the action space;

- $\mathcal{P}(s'|s, a)$ denotes the transition dynamics where $s, s' \in \mathcal{S}$ and $a \in \mathcal{A}$;

- $\mathcal{R} = [r_1, \ldots, r_m]^T$ is a vector of $m$ reward functions, each $r_i : \mathcal{S} \times \mathcal{A} \to \mathbb{R}$;

- $\boldsymbol{\gamma} = [\gamma_1, \ldots, \gamma_m]^T$ are per-objective discount factors;

- $\mathcal{D}$ is the initial state distribution.

The goal is to learn a stochastic policy $\pi_\theta(a|s)$ that maximizes a vector of expected returns:

$$\boldsymbol{J}^\pi = [J_1^\pi, \ldots, J_m^\pi]^T, \text{ where } J_i^\pi = \mathbb{E}_{\pi, \mathcal{P}} \left[ \sum_{t=0}^T \gamma_i^t r_i(s_t, a_t) \right].$$

Unlike standard RL, no single policy typically maximizes all objectives simultaneously, motivating the use of multi-objective optimization.

### 2.2. Multi-Objective Optimization

In multi-objective learning, we aim to optimize the vector-valued objective:

$$\max_\pi \boldsymbol{F}(\pi) = [f_1(\pi), f_2(\pi), \ldots, f_m(\pi)], \quad (1)$$

where each $f_i(\pi) = J_i^\pi$ corresponds to an expected return under objective $i$. As objectives may conflict, the solution is characterized by the Pareto front.

**Definition 2.1.** *(Pareto Optimality)* A policy $\pi$ is said to *dominate* another policy $\pi'$ if:

$$\boldsymbol{F}(\pi) \succeq \boldsymbol{F}(\pi') \quad \text{and} \quad \boldsymbol{F}(\pi) \neq \boldsymbol{F}(\pi'). \quad (2)$$

A policy is *Pareto-optimal* if it is not dominated by any other policy. The set of such policies forms the *Pareto set*, and their image in objective space is the *Pareto front*.

In multi-objective learning, the true Pareto-optimal set is typically intractable due to the complexity of the objective space. Instead, optimization methods aim to approximate this front with a finite set of nondominated solutions. To evaluate such approximations, the hypervolume metric (Zitzler & Thiele, 1998) is widely used, as it jointly reflects both convergence and diversity by measuring the volume dominated by the solution set relative to a reference point.

**Definition 2.2.** *(Hypervolume)* The hypervolume (HV) is a standard quality indicator of Pareto approximations. Given a set of non-dominated outcomes $P$ and a reference point $\boldsymbol{r}$, the hypervolume is defined as the dominated volume:

$$\text{HV}(P) = \int_{\mathbb{R}^m} \mathbb{1}_{\boldsymbol{r} \preceq \boldsymbol{z} \preceq \boldsymbol{p}, \, \exists \boldsymbol{p} \in P} \, d\boldsymbol{z}, \quad (3)$$

where $\preceq$ is the relation operator of objective dominance. A larger HV implies better coverage and spread along the Pareto frontier (Zitzler & Thiele, 1998).

### 2.3. Problem Formulation

We formulate the training of a tool-using LLM agent as a multi-objective MDP. At each time step, the agent chooses an action which could be either a tool API call or a direct output token. Upon episode completion, the agent receives a vector reward $\boldsymbol{r}_\theta = (r_{task}, r_{tool})$: in our case, $r_{task}$ measures task performance (e.g., accuracy), and $r_{tool}$ reflects tool-use efficiency, which is defined as:

$$r_{tool} = \exp(-\alpha |N_{call} - N_{optimal}|), \quad (4)$$

where $N_{call}$ denotes the actual number of tool calls in a given trajectory, $N_{optimal}$ is the estimated optimal number of calls for a task $q$, and $\alpha$ is a hyper-parameter controlling the penalty severity. Note that we calculate the minimal tool calls $N_{optimal} = \min(\mathcal{C})$, where $\mathcal{C} = \{c_1, c_2, ..., c_k\}$

is the number of tool calls only for current successful trajectories $\{\tau_1, \tau_2, ..., \tau_k\}$ and $N_{call} \geq N_{optimal}$. This way serves as the local approximation of optimal tool calls for that task (Wang et al., 2025a). Moreover, we track and update $N_{optimal}$ during multi-epoch training to approximate global optimal tool calls if the agent can find a better solution in the later training. The agent's objective is to maximize both $r_{task}$ and $r_{tool}$ over its policy $\pi_\theta$. We call an outcome $(r_{task}, r_{tool})$ *Pareto-optimal* if no other policy achieves higher $r_{task}$ and higher $r_{tool}$ simultaneously.

## 3. Approach

In this section, we present our main contribution **ParetoPO**, a two-stage reinforcement learning framework for training tool-integrated agents under multiple conflicting objectives. This approach combines dynamic scalarization with dominance-aware policy optimization to encourage exploration of the Pareto front and enable fine-grained behavior refinement without fixed reward weights. Algorithm 1 presents the overview of our method. The full details and theoretical analysis are given in Sections 3.1-3.3.

### 3.1. Hypervolume-Guided Dynamic Scalarization

Previous work (Yao et al., 2025) maintains a static weight vector $\boldsymbol{w} = (w_1, w_2)$ for scalarizing the vector reward into a scalar reward as $r_w = \boldsymbol{w}^\top \boldsymbol{r}$. To address their limitations, we adopt a hypervolume-guided dynamic scalarization method, which adaptively adjusts the effective reward weights based on smoothed hypervolume signal.

**Meta-Level Reward Weight Adaptation.** Given the current Pareto set $\boldsymbol{B} = \{\boldsymbol{r}_{\theta_0}, ..., \boldsymbol{r}_{\theta_{t-1}}\}$ achieved so far and a new outcome vector $\boldsymbol{r}$, we calculate the *hypervolume contribution* for this outcome as $\Delta\mathrm{HV}(\boldsymbol{r}, \boldsymbol{B}) = \mathrm{HV}(\boldsymbol{B} \cup \boldsymbol{r}) - \mathrm{HV}(\boldsymbol{B})$, using the recursive dimension sweep algorithm in Fonseca et al. (2006). In prior work (Lu et al., 2025), they do not directly adjust the reward weights $\boldsymbol{w}$ but instead adopt a meta-level reward $r_{pareto}$ that relies on detecting immediate positive hypervolume contributions $\Delta\mathrm{HV}$ as a trigger for reward amplification. However, in tool-use scenarios, outcomes can be noisy or highly variable, e.g., using a tool might occasionally produce a big jump in one objective but at the cost of others, or there may be stochastic tool failures. A naive hypervolume signal might be unstable, either over-emphasizing one lucky improvement or providing zero reward during noisy phases. Therefore, we adopt **smoothed hypervolume gain** as metric at the $t$-th training step as:

$$\Delta\overline{\mathrm{HV}}_t = \gamma\Delta\overline{\mathrm{HV}}_{t-1} + (1 - \gamma)\Delta\mathrm{HV}_t, \qquad (5)$$

where $\gamma$ is a smoothing factor. This smoothing dampens the effect of outlier episodes and provides a more stable learning signal. Thus, the meta-level reward $r_{pareto}$ is defined as

a monotonic function of the smoothed gain and the final reward is adjusted via $r_{pareto}$:

$$
\begin{aligned}
r_{pareto}(\boldsymbol{r}, \boldsymbol{B}) &= 0.5 + 1.5\tanh(\Delta\overline{\mathrm{HV}}_t(\boldsymbol{r}, \boldsymbol{B})), \\
\tilde{r}_w &= r_{pareto} \cdot r_w.
\end{aligned} \qquad (6)
$$

Importantly, since $r_{pareto}$ depends on the current Pareto archive $\boldsymbol{B}$, which evolves as new non-dominated outcomes are discovered, the resulting scalarized return $\tilde{r}_w = r_{pareto} \cdot \boldsymbol{w}^\top \boldsymbol{r}$ induces a time-varying effective objective. This mechanism implicitly reweights trade-offs among objectives, encouraging exploration of under-represented regions of the Pareto front.

**Iterative Weights Adaptation.** At each training step $t$, we update the policy $\pi_\theta$ using a batch of trajectories and re-evaluate its performance on a held-out validation set. Given a training batch $\mathcal{D}_b$, the agent samples $g$ responses for each query $q \in \mathcal{D}_b$ under the current policy $\pi_{\theta_{t-1}}$. Each trajectory $\tau_i$ is first assigned a vector reward $\boldsymbol{r} = (r_{task}, r_{tool})$. We then compute the scalarized reward $r_w = \boldsymbol{w}^\top \boldsymbol{r}$ and rescale it via the meta-level amplification term defined in Eq. 6. The policy parameters are then optimized using these augmented scalar rewards via GRPO (Shao et al., 2024). After the update, the Pareto set $\boldsymbol{B}$ (the archive of best-so-far objective vectors) is updated to include any newly discovered non-dominated outcomes.

### 3.2. Pareto Ranking Policy Optimization

Even with dynamic scalarization, the policy update in standard RL still relies on scalar returns that may bias learning toward *a particular weight setting*. To more directly align the policy optimization with multi-objective trade-offs, we propose a Pareto ranking based advantage computation mechanism. The key idea is to compute advantage values by comparing each trajectory's outcome against others in a batch using Pareto dominance.

**Pareto Ranking for Trajectory Evaluation.** We leverage the concept of Pareto dominance to evaluate and rank trajectories in a multi-objective setting. Given a group of rollout trajectories, a trajectory $\tau_i$ *dominates* another trajectory $\tau_j$ if $\tau_i$ is no worse than $\tau_j$ on all objectives and strictly better on at least one. A trajectory is **Pareto optimal** (nondominated) within the batch if no other trajectory dominates it. Using this criterion, we perform non-dominated sorting to assign each trajectory a Pareto rank $\rho$, where rank 1 contains all nondominated trajectories, rank 2 contains those dominated only by rank-1 trajectories, and so on. This Pareto ranking provides an ordinal assessment of each trajectory's quality in the multi-objective sense.

**Advantage Assignment via Pareto Rank.** Once trajectories are ranked, we compute their advantage values based

on their ranks. We define a base advantage for rank $\rho$ as:

$$A_{base,\rho} = N_{rank} - \rho + 1, \qquad (7)$$

where $N_{rank}$ is the total number of ranks. To reflect varying preference over objectives across tasks, we further refine advantages within each rank by incorporating the trajectory's outcome $\boldsymbol{r} = (r_{task}, r_{tool})$ and the user-defined weight vector $\boldsymbol{w} = (w_1, w_2)$. Specifically, we first define a normalized reward for trajectory $\tau_i$ in rank $\rho$ as:

$$\hat{r}_w = \frac{r_w - r_{min}}{r_{max} - r_{min}}, \qquad (8)$$

where $r_w = \boldsymbol{w}^\top \boldsymbol{r}$, and $r_{min}, r_{max}$ are the minimum and maximum rewards within the rank $\rho$, respectively. When $r_{min} = r_{max}$, $\hat{r}_w = 0.5$. Finally, we adjust the trajectory's advantage $A_i$ based on this normalized score as:

$$A_i = A_{base,\rho} + \beta \cdot (\hat{r}_w - 0.5), \qquad (9)$$

where $\beta \leq 1$ is a scaling factor. Notably, this formulation guarantees that no trajectory from a worse rank can receive a higher advantage than any trajectory from a better rank. This step injects the preference priorities into the advantage values, so that those trajectories that perform better on the higher-priority objective (according to the weights) get a slight boost in advantage. Based on the refined advantage, we adopt GRPO to update the policy.

**Discussion of Training Stability and Overhead.** The efficiency reward $r_{tool}$ depends on a moving $N_{optimal}$, which may introduce some non-stationarity. However, $N_{optimal}$ is defined as the minimum tool count among successful trajectories, so it is monotonically non-increasing over training. Unlike a noisy reward baseline, it changes only when a more efficient successful trajectory is found, which makes its dynamics stable. In addition, the efficiency reward is bounded in $[0, 1]$, preventing unstable reward magnitudes. Besides, as training progresses and the policy improves, discovering trajectories with even fewer tool calls becomes increasingly rare. As a result, $N_{optimal}$ quickly stabilizes and the optimization becomes approximately stationary in later stages. In our approach, Pareto ranking is performed within a small group of sampled rollouts for each query. Therefore, even if the proportion of non-dominated trajectories increases with more objectives, this effect is bounded within a very small candidate set and it does not lead to an uncontrolled solution explosion in ranking cost.

### 3.3. Theoretical Analysis

Figure 1 illustrates trajectories of the two stages in multi-objective optimization. In our method, Stage 1 adopts a dynamic scalarization strategy to generate a diverse set of policies with different trade-offs among objectives. By

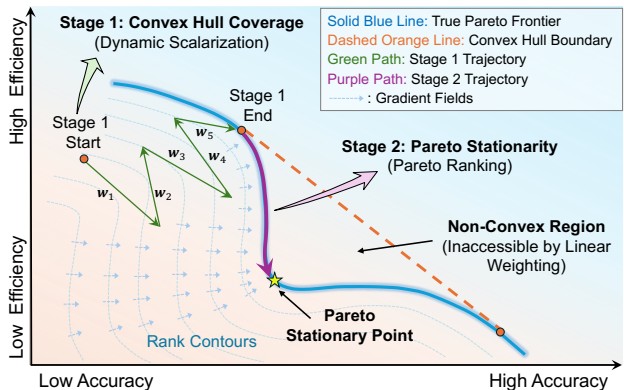

*Figure 1.* Two-stage multi-objective RL: Accuracy vs. Efficiency.

repeatedly optimizing scalarized objectives and retaining non-dominated outcomes, Stage 1 expands the coverage of achievable returns in the objective space, providing a global approximation of the Pareto front (stated in Proposition 3.1).

**Proposition 3.1 (Supported-Hull Coverage of Stage 1).** *Let $\mathcal{Y} = \{\boldsymbol{J}(\pi) : \pi \in \Pi\}$ denote the set of achievable expected return vectors, and let $\mathcal{S}_T = \{\boldsymbol{J}(\pi_t)\}_{t=1}^T$ be the set of evaluation vectors generated in Stage 1. Assume that dynamic scalarization explores preference directions densely over the simplex and approximately optimizes each visited scalarization. Then the discovered hull $\mathcal{C}_T = \mathrm{conv}(\mathcal{S}_T)$ converges to the achievable hull $\mathcal{C} = \mathrm{conv}(\mathcal{Y})$ in support-function distance over the simplex:*

$$\sup_{\boldsymbol{w} \in \Delta^M} \left| h_{\mathcal{C}}(\boldsymbol{w}) - h_{\mathcal{C}_T}(\boldsymbol{w}) \right| \to 0 \qquad (T \to \infty), \quad (10)$$

*where $h_{\mathcal{C}}(\boldsymbol{w}) = \sup_{\boldsymbol{y} \in \mathcal{C}} \boldsymbol{w}^\top \boldsymbol{y}$. Specifically, Stage 1 asymptotically covers all supported Pareto-optimal points, i.e., all points $\boldsymbol{y}^* \in \mathcal{Y}$ that maximize $\boldsymbol{w}^\top \boldsymbol{y}$ for some $\boldsymbol{w} \in \Delta^M$.*

*Proof.* Appendix C. □

To analyze Stage 2, the main difficulty is that Pareto ranks obtained by non-dominated sorting are discrete and non-differentiable. Thus, we introduce a stochastic smoothing of Pareto ranks by adding i.i.d. Gumbel noise.

**Definition 3.2 (Stochastic Pareto Rank (Order-Consistent Surrogate)).** For each trajectory $\tau$, let $\tilde{\boldsymbol{G}}(\tau) = \boldsymbol{G}(\tau) + \xi$ where $\xi_m \sim \mathrm{Gumbel}(0, \sigma)$ i.i.d. For two trajectories $\tau_i, \tau_j$, define the stochastic dominance probability

$$p_\sigma(\tau_i \succ \tau_j) \triangleq \mathbb{P}\Big(\tilde{\boldsymbol{G}}(\tau_i) \succeq \tilde{\boldsymbol{G}}(\tau_j)\Big),$$

where the probability is taken over the Gumbel noise. We define the stochastic Pareto rank surrogate of $\tau_i$ (lower is better) as $R_\sigma(\tau_i) \triangleq 1 + \sum_{j \neq i} p_\sigma(\tau_j \succ \tau_i)$. For any fixed $\sigma > 0$, $R_\sigma(\tau_i)$ is bounded and is a continuous (almost-everywhere differentiable) function of the perturbed returns.

As $\sigma \to 0$, the surrogate becomes increasingly sharp and is order-consistent with Pareto dominance: if $\tau_i \succ_P \tau_j$, then $R_\sigma(\tau_i) < R_\sigma(\tau_j)$ with probability approaching 1.

**Assumption 3.3** (**Monotonicity of Advantage**). *The shaping function $\Phi(r, \hat{r}_w)$ is monotone decreasing in its rank argument $r$. In particular, if $\tau_i \succ_P \tau_j$, then $\Phi(R_\sigma(\tau_i), \cdot) > \Phi(R_\sigma(\tau_j), \cdot)$ for $\sigma$ sufficiently small.*

**Lemma 3.4** (**Expected Batch Gradient as a Pareto-Ascent Direction**). *Let $A_\sigma(\tau_i; \tau_{-i})$ be the smoothed rank-based advantage of trajectory $\tau_i$ computed within a batch $\tau_{1:K}$, with $\sigma > 0$ and shaping function $\Phi$ satisfying Assumption 3.3. Define the batch-level smoothed objective*

$$W_{\sigma,K}(\theta) = \mathbb{E}_{\tau_{1:K} \sim \pi_\theta, \xi} \left[ \frac{1}{K} \sum_{i=1}^{K} A_\sigma(\tau_i; \tau_{-i}) \right],$$

*and its mean-field gradient*

$$\boldsymbol{g}_{\sigma,K}(\theta) \triangleq \nabla_\theta W_{\sigma,K}(\theta).$$

*Then $\boldsymbol{g}_{\sigma,K}(\theta)$ is a Pareto-ascent direction in the following sense: for any direction $d$ such that $\nabla_\theta J_m(\theta)^\top d \geq 0$ for all $m \in \{1, \ldots, M\}$, we have $\boldsymbol{g}_{\sigma,K}(\theta)^\top d \geq 0$. Moreover, as $\sigma \to 0$, $A_\sigma(\tau_i; \tau_{-i}) \to A(\tau_i; \tau_{-i})$ almost surely, and $\boldsymbol{g}_{\sigma,K}(\theta)$ converges to the corresponding unsmoothed batch gradient by dominated convergence.*

*Proof.* Appendix D. □

**Theorem 3.5** (**Bounded Second Moment of Rank-Based Policy-Gradient Estimators**). *The rank-based advantage is uniformly bounded:*

$$|A(\tau_i; \tau_{-i})| \leq N_{rank} + \beta/2,$$

*for any trajectory $\tau_i$ within a batch $\tau_{1:K}$. Define the per-trajectory policy-gradient estimator within a batch as*

$$\hat{\boldsymbol{g}}(\theta; \tau_i) \triangleq A(\tau_i; \tau_{-i}) \nabla_\theta \log \pi_\theta(\tau_i), \qquad \tau_{1:K} \sim \pi_\theta.$$

*Then its second moment is bounded as*

$$\mathbb{E}\left[ \|\hat{\boldsymbol{g}}(\theta; \tau_i)\|^2 \right] \leq C \cdot \mathbb{E}\left[ \|\nabla_\theta \log \pi_\theta(\tau_i)\|^2 \right], \quad (11)$$

*for a constant $C = (N_{rank} + \beta/2)^2$ independent of the reward magnitudes. As a result, the minibatch estimator $\hat{\boldsymbol{g}}_b(\theta) = \frac{1}{K} \sum_{i=1}^{K} \hat{\boldsymbol{g}}(\theta; \tau_i)$ has a bounded second moment.*

*Proof.* Appendix E. □

**Theorem 3.6** (**Convergence to Pareto-Ascent Stationarity of Stage 2**). *Let the policy parameters evolve according to the stochastic approximation $\theta_{t+1} = \theta_t + \eta_t \hat{\boldsymbol{g}}_b(\theta_t)$, so*

$$\hat{\boldsymbol{g}}_b(\theta_t) = \frac{1}{K} \sum_{i=1}^{K} A(\tau_i; \tau_{-i}) \nabla_\theta \log \pi_{\theta_t}(\tau_i),$$

*where $\tau_{1:K} \sim \pi_{\theta_t}$ and $\{\eta_t\}$ satisfies the Robbins-Monro conditions $\sum_t \eta_t = \infty$ and $\sum_t \eta_t^2 < \infty$. Assume each objective return $J_m(\theta)$ is continuously differentiable with Lipschitz gradient. Define the batch mean-field vector field*

$$\boldsymbol{g}_{\sigma,K}(\theta) \triangleq \mathbb{E}_{\tau_{1:K} \sim \pi_\theta}[\hat{\boldsymbol{g}}_b(\theta)] = \nabla_\theta W_{\sigma,K}(\theta),$$

*where $W_{\sigma,K}$ is the smoothed batch objective defined in Lemma 3.4. Then every limit point $\theta^\star$ of $\{\theta_t\}$ satisfies the following Pareto-ascent stationarity condition:*

$$\nabla_\theta J_m(\theta^\star)^\top d \geq 0 \ \forall m \implies \boldsymbol{g}_{\sigma,K}(\theta^\star)^\top d = 0. \quad (12)$$

*Equivalently, there exists no direction $d$ that simultaneously improves all objectives to first order and yields a strictly positive expected policy update. In this sense, $\theta^\star$ is stationary with respect to all Pareto-ascent directions.*

*Proof.* Appendix F. □

*Remark* 3.7 (Why Two Stages?). The two-stage structure of our method admits a principled theoretical interpretation. Stage 1 leverages hypervolume-guided dynamic scalarization to explore and approximate the supported boundary of the achievable return set, providing global coverage of diverse linear trade-offs among objectives (Proposition 3.1). Stage 2 then performs local policy improvement using Pareto ranking based advantages, and converges to Pareto-ascent stationary points that admit no common first-order improvement direction for all objectives (Theorem 3.6). This separation allows Stage 1 to address global exploration of trade-offs, while Stage 2 ensures locally Pareto-stable refinement without relying on fixed scalarization weights.

# 4. Experiments

## 4.1. Experimental Setup

**Datasets and Metrics.** We conduct experiments on two categories of complex reasoning tasks that require external tools: *mathematical reasoning* and *multi-hop QA*. For math reasoning, we employ five benchmark datasets: **MATH500** (Hendrycks et al., 2021), **AIME2024** (MAA, 2024), **AIME2025** (MAA, 2025), **OlympiadBench** (He et al., 2024), and **AMC23** (MAA, 2023), covering a wide range of competition-level arithmetic and symbolic problems. These tasks require multi-step logical reasoning and invoke Python interpreter as an external tool. For multi-hop QA tasks, we adopt the **Natural Questions (NQ)** (Kwiatkowski et al., 2019) and **HotpotQA** (Yang et al., 2018) datasets, which involve open-domain question answering and require the agent to interact with a retrieval system for evidence lookup. To evaluate model performance, we report two metrics across all tasks: (1) *Exact Match (EM)*, which measures the final answer accuracy, and (2)

*Table 1.* Evaluation results on five datasets on mathematical reasoning. "EM" and "#Tool" denote exact match and the average number of tool calling, respectively. The **bold** and underline fonts denote the best and second best results in each dataset, respectively. "*" indicates the statistically significant improvements (i.e., t-test with $p < 0.05$) over the best baseline.

| Model | Tool Use | MATH500 | | AIME24 | | AIME25 | | Olympiad | | AMC23 | |
|---|---|---|---|---|---|---|---|---|---|---|---|
| | | EM | #Tool | EM | #Tool | EM | #Tool | EM | #Tool | EM | #Tool |
| Qwen2.5-Math-1.5B | | | | | | | | | | | |
| Qwen2.5-Math-1.5B-Ins | No | 66.0 | / | 10.0 | / | 10.0 | / | 31.0 | / | 62.5 | / |
| Qwen2.5-Math-1.5B-Ins-TIR | Yes | 73.8 | 1.3 | 13.3 | 1.1 | 13.3 | 1.4 | 41.3 | 1.5 | 55.0 | 2.0 |
| ToRL-GRPO | Yes | 77.8 | 2.1 | 23.3 | 2.2 | 23.3 | 2.3 | 44.0 | 2.7 | 67.5 | 2.5 |
| OTC-GRPO | Yes | 74.0 | 1.3 | 20.0 | 1.1 | 20.0 | 1.1 | 42.1 | 1.2 | 62.5 | 1.1 |
| MO-GRPO | Yes | 71.2 | 2.0 | 16.7 | 1.8 | 16.7 | 1.8 | 41.2 | 2.0 | 62.5 | 2.1 |
| ParetoPO (Ours) | Yes | **80.0*** | **0.9** | **30.0*** | **0.8** | **30.0*** | **0.8** | **48.1*** | **0.8** | **70.0*** | **0.8** |
| Qwen2.5-Math-7B | | | | | | | | | | | |
| Qwen2.5-Math-7B-Ins | No | 74.8 | / | 10.0 | / | 16.7 | / | 32.4 | / | 65.0 | / |
| Qwen2.5-Math-7B-Ins-TIR | Yes | 78.8 | 1.6 | 26.7 | 1.6 | 16.7 | 1.4 | 45.0 | 1.8 | 70.0 | 2.3 |
| ToRL-GRPO | Yes | 82.2 | 2.2 | **36.7** | 2.1 | 26.7 | 2.1 | 49.9 | 2.8 | 75.0 | 2.6 |
| OTC-GRPO | Yes | 81.0 | 1.5 | **36.7** | **0.7** | 23.3 | **0.8** | 46.1 | 1.2 | 72.5 | 1.6 |
| MO-GRPO | Yes | 80.8 | 1.9 | 26.7 | 1.8 | 23.3 | 1.9 | 46.3 | 1.7 | 67.5 | 1.7 |
| ParetoPO (Ours) | Yes | **84.6*** | **1.2** | **36.7** | 0.8 | **33.3*** | **0.8** | **52.2*** | **0.9** | **77.5*** | **1.3** |

*Tool Usage Count (#Tool)*, which records the number of external tool calls made per trajectory. This dual evaluation captures both task success and reasoning efficiency, highlighting the effectiveness of multi-objective optimization in controlling tool reliance while preserving accuracy. The details of datasets are detailed in Appendix A.

**Baselines.** We compare our method against a suite of baselines covering non-tool, tool-integrated, and multi-objective optimization settings. For *mathematical reasoning* tasks, we compare to (1) **Qwen2.5-Math-1.5B-Instruct** (Yang et al., 2024) without tool use; (2) **Qwen2.5-Math-1.5B-Instruct-TIR** (Yang et al., 2024), a tool-integrated reasoning (TIR) model that invokes Python interpreter as tool; (3) **ToRL-GRPO** (Li et al., 2025c), a tool-using agent trained with GRPO to maximize task success; (4) **OTC-GRPO** (Wang et al., 2025b) and (5) **MO-GRPO** (Ichihara et al., 2025), two GRPO-trained baselines that consider both task accuracy and tool efficiency using different reward weighting strategies. For *multi-hop QA* tasks, we compare with: (1) **R1-Base**, a base model fine-tuned with reinforcement learning without tool usage; (2) **SFT**, a supervised fine-tuned model trained on gold answers; (3) **RAG**, a retrieval-augmented generation model that calls the retriever to search evidence; and (4) three GRPO-based tool-integrated models: **Search-R1** (Jin et al., 2025), **OTC-GRPO** (Wang et al., 2025b), and **MO-GRPO** (Ichihara et al., 2025), each representing different reward formulations or optimization strategies for balancing accuracy and retrieval efficiency. This comprehensive suite enables a detailed analysis of optimization objectives, and their impact on performance and efficiency. The implementation details are shown in Appendix B.

*Table 2.* Evaluation results on two datasets on multi-hop QA.

| Model | Tool Use | NQ | | HotpotQA | |
|---|---|---|---|---|---|
| | | EM | #Tool | EM | #Tool |
| Qwen2.5-3B (Base) | | | | | |
| R1-Base | No | 22.6 | / | 20.1 | / |
| SFT | No | 24.9 | / | 18.6 | / |
| RAG | Yes | 34.8 | 1.0 | 25.5 | 1.0 |
| Search-R1 | Yes | 40.4 | 1.4 | 31.2 | 1.8 |
| OTC-GRPO | Yes | 44.4 | 1.0 | 36.5 | 1.4 |
| MO-GRPO | Yes | 41.2 | 1.8 | 32.0 | 1.9 |
| ParetoPO (Ours) | Yes | **48.0*** | **0.9** | **42.2*** | **1.2** |
| Qwen2.5-7B (Base) | | | | | |
| R1-Base | No | 27.0 | / | 24.2 | / |
| SFT | No | 31.8 | / | 21.7 | / |
| RAG | Yes | 34.9 | 1.0 | 29.9 | 1.0 |
| Search-R1 | Yes | 39.9 | 1.7 | 34.1 | 2.1 |
| OTC-GRPO | Yes | 44.4 | 1.0 | 36.6 | 1.0 |
| MO-GRPO | Yes | 41.5 | 1.4 | 33.1 | 1.6 |
| ParetoPO (Ours) | Yes | **50.1*** | **0.9** | **45.2*** | **0.9** |

### 4.2. Main Results

Tables 1 and 2 summarize the results on mathematical reasoning and multi-hop QA tasks, respectively. Across both settings, ParetoPO consistently achieves the best trade-off between task performance and tool-use efficiency.

On mathematical reasoning benchmarks, methods that optimize task accuracy alone (e.g., ToRL-GRPO) achieve strong EM scores but rely heavily on tool usage, while methods that explicitly penalize tool calls (e.g., OTC-GRPO) reduce tool usage at the cost of noticeable accuracy degradation. MO-GRPO exhibits similar trade-off behavior but lacks consistency across datasets. In contrast, ParetoPO attains the highest EM across most datasets and two model sizes while maintaining the lowest average number of tool calls,

*Table 3.* Ablation study in MATH500 and NQ.

| Method | MATH500 | | NQ | |
|---|---|---|---|---|
| | EM | #Tool | EM | #Tool |
| ParetoPO | 80.0 | 0.9 | 48.0 | 0.9 |
| w/o Stage 1 | 78.5 | 0.9 | 47.0 | 0.9 |
| w/o Stage 2 | 76.3 | 1.1 | 46.1 | 1.0 |
| w/o Dynamic Weighting | 79.1 | 0.9 | 47.3 | 0.9 |
| w/o Pareto Ranking | 77.0 | 1.0 | 46.9 | 1.0 |

*Table 4.* Analysis of pref-defined weights in scalarization.

| $(w_1, w_2)$ | MATH500 | | NQ | |
|---|---|---|---|---|
| | EM | #Tool | EM | #Tool |
| (0.2, 0.8) | 71.5 | 0.6 | 41.3 | 0.6 |
| (0.8, 0.2) | 78.7 | 1.1 | 47.6 | 1.0 |
| (0.4, 0.6) | 78.2 | 0.8 | 47.5 | 0.8 |
| (0.6, 0.4) | 80.0 | 0.9 | 48.0 | 0.9 |
| (0.5, 0.5) | 79.2 | 0.9 | 47.6 | 0.8 |

demonstrating its ability to navigate multi-objective trade-offs without sacrificing either objective.

A similar pattern is observed on other multi-hop QA tasks. ToRL-GRPO again favors accuracy with higher tool usage, whereas OTC-GRPO reduces tool calls but underperforms in EM. ParetoPO consistently achieves the best accuracy across datasets and model scales, while using substantially fewer tool calls. These results validate the effectiveness of our two-stage optimization framework, which avoids fixed reward compositions and instead leverages hypervolume-guided dynamic scalarization and Pareto-ranking-based policy optimization to balance conflicting objectives.

In Table 1 and 2, our approach achieves both higher accuracy and higher efficiency than baselines. These results suggest that many single-objective methods might be not on a strong Pareto frontier. In tool-using agents, excessive tool use can introduce noise, lengthen trajectories, and worsen credit assignment, so reducing redundant tool calls can improve both efficiency and accuracy. Theoretically, our method explicitly optimizes multiple objectives jointly: Stage 1 explores supported Pareto regions via dynamic scalarization, and Stage 2 uses Pareto ranking to favor nondominated trajectories and refine the policy toward Pareto-ascent stationary points. This helps ParetoPO move away from process-inefficient local optima toward a better Pareto region.

### 4.3. Further Analysis

#### 4.3.1. ABLATION ANALYSIS

To assess the contribution of each component, we conduct ablation studies on MATH500 and NQ using Qwen2.5-Math-1.5B and Qwen2.5-3B (Base), respectively. As shown in Table 3, we evaluate four variants of ParetoPO by removing or modifying key components.

Across both tasks, all ablated variants underperform the full model, confirming that each component contributes to the overall effectiveness. Removing Stage 1 (dynamic scalarization) leads to a consistent drop in EM, indicating that adaptive scalarization plays an important role in exploring diverse Pareto trade-offs. In contrast, removing Stage 2 (Pareto-ranking-based optimization) results in a larger performance degradation, particularly in tool efficiency, sug-

gesting that dominance-based advantage estimation is critical for refining policy behavior and controlling efficiency.

Replacing dynamic scalarization with fixed-weight scalarization (w/o Dynamic Weighting, $w = (0.6, 0.4)$) further degrades both accuracy and tool efficiency, highlighting the limitations of static linear reward compositions. Similarly, disabling Pareto ranking and using GRPO-style advantage estimation (w/o Pareto Ranking) leads to noticeable drops in both objectives, indicating that scalarized returns alone are insufficient to capture local Pareto trade-offs.

Overall, these results show that two stages are complementary: dynamic scalarization facilitates global exploration of the objective space, while Pareto-ranking-based optimization enables dominance-aware refinement, and both are necessary for achieving strong multi-objective performance.

#### 4.3.2. TRAINING DYNAMICS ANALYSIS

To better understand how different optimization objectives shape agent behavior during learning, we analyze training dynamics on NQ using Qwen2.5-3B (Base). Figure 2 tracks the evolution of three quantities over training steps: number of search calls, exact match (EM), and response length.

As shown in Figure 2 (Left), Search-R1 increasingly relies on external search, with the average number of search calls rising from about 1.5 to over 2.5. In contrast, ParetoPO steadily reduces tool usage and converges to around 0.9 calls, while OTC-GRPO exhibits an initial decrease followed by oscillations around 1.0. Overall, ParetoPO achieves the lowest tool usage, indicating that the Pareto-style objective effectively discourages unnecessary search.

Figure 2 (Middle) shows that all methods improve in EM during training, but ParetoPO achieves faster gains and a higher final plateau. Notably, reduced tool usage under ParetoPO does not compromise accuracy; instead, the policy becomes both more efficient and more effective.

Finally, Figure 2 (Right) highlights clear differences in generation behavior. Search-R1 produces increasingly long responses as training progresses, suggesting a tendency toward verbose reasoning and higher inference cost. ParetoPO, by contrast, steadily shortens its responses and stabilizes around 650 tokens. This indicates that ParetoPO mitigates length inflation and promotes more concise gener-

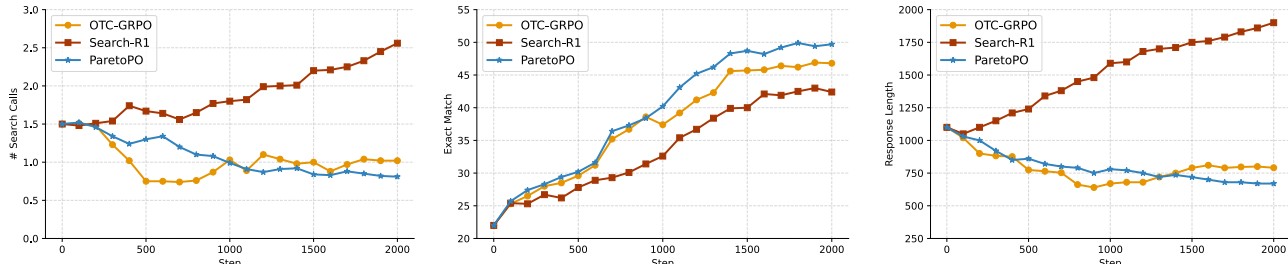

*Figure 2.* **Left**: Changes of number of search calls during the training; **Middle**: Changes of exact match (EM) score during the training; and **Right**: Changes of response length during the training.

ation alongside improved accuracy.

### 4.3.3. HYPER-PARAMETER ANALYSIS

To evaluate the robustness of ParetoPO to the choice of scalarization weights, we conduct a sensitivity analysis by varying $(w_1, w_2)$, where $w_1$ emphasizes task accuracy and $w_2$ emphasizes tool efficiency. We evaluate on MATH500 with Qwen2.5-Math-1.5B and on NQ with Qwen2.5-3B (Base); results are summarized in Table 4.

Across both settings, performance remains stable over a broad range of weight choices. As expected, increasing $w_2$ leads to fewer tool calls at the cost of lower EM, while shifting emphasis toward accuracy improves EM with only a moderate increase in tool usage. Notably, the best accuracy is consistently achieved by moderately accuracy-biased but non-extreme weights. For example, $(w_1, w_2) = (0.6, 0.4)$ yields the highest EM on both MATH500 and NQ while maintaining low tool usage, and balanced weights such as $(0.5, 0.5)$ perform competitively with similar efficiency.

Overall, these results indicate that ParetoPO is robust to the choice of scalarization weights and does not rely on narrow hyperparameter tuning. A moderate emphasis on accuracy (e.g., $w_1 \in [0.5, 0.6]$) provides a strong default choice.

### 4.3.4. PARETO FRONTIERS ANALYSIS

Figure 3 visualizes the Pareto frontiers between task accuracy ($r_{task}$) and tool efficiency ($r_{tool}$, Eq. (4)) for ParetoPO and competing baselines. Each point on the curves corresponds to a model checkpoint at a different training step, reflecting how the trade-off evolves during optimization.

Across both benchmarks, ParetoPO traces a substantially broader and smoother Pareto frontier over training. In the early and intermediate stages, ParetoPO naturally explores efficiency-focused solutions with high tool efficiency while maintaining competitive task accuracy. As training progresses, it transitions smoothly toward balanced operating points, achieving strong task accuracy with only moderate efficiency loss. In contrast, baseline methods exhibit more polarized behavior: OTC-GRPO remains largely confined

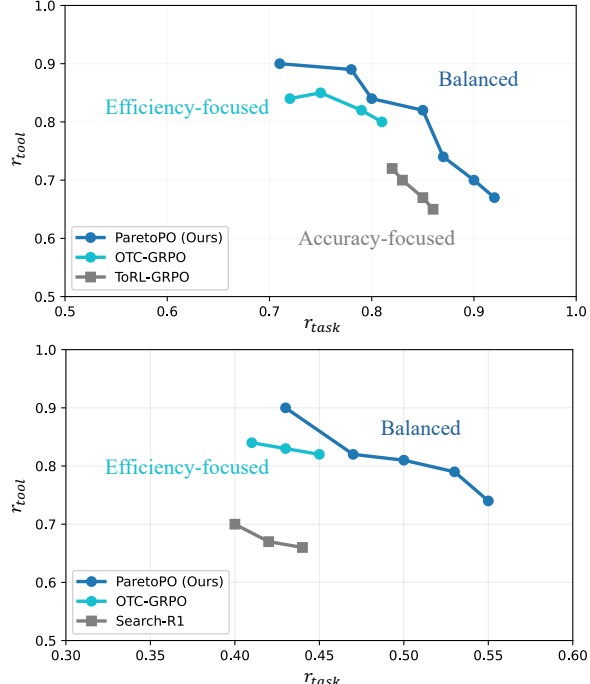

*Figure 3.* **Top**: Qwen2.5-Math-7B on Math500; **Bottom**: Qwen2.5-7B (Base) on Natural Questions.

to efficiency-oriented regions, while accuracy-focused baselines (e.g., ToRL-GRPO) improve task accuracy at the cost of a pronounced drop in tool efficiency.

## 5. Related Work

**Multi-Objective Reinforcement Learning.** Multi-objective RL addresses problems with multiple conflicting criteria, seeking policies that achieve a desirable trade-off among objectives. Early work (Lu et al., 2025; Li et al., 2025a) established that optimizing each objective independently typically produces very different policies, necessitating methods to handle simultaneous optimization. A variety of MORL techniques have been proposed, including scalarization approaches (Li et al., 2025b), post-hoc policy mixing (Rame et al., 2023; Wang et al., 2025c), and multi-policy

architectures (Callaghan et al., 2025). A common approach is linear scalarization of rewards, but it is well-known that any fixed linear weight vector can miss Pareto-optimal solutions on concave regions of the Pareto front (Lu et al., 2025). To capture the full Pareto set, some methods (Wang et al., 2024a; Zhong et al., 2024) train a set of policies, each for a different weight or preference setting (e.g., via continuation methods or preference-conditioned policies). While multi-policy approaches (including evolutionary algorithms) can approximate the entire Pareto front (Callaghan et al., 2025), they incur significant computational cost by maintaining many policies or a large population of solutions.

**LLM Alignment and Efficiency.** Recently, there has been growing interest in aligning LLMs to multiple objectives beyond pure accuracy or reward scores. Several works have integrated secondary objectives such as response length (Cheng et al., 2025), token efficiency (Zhang et al., 2025), factuality (Li & Ng, 2025), or safety (Li et al., 2025b) into the training of LLMs. For example, on-policy methods like PPO (Schulman et al., 2017) or GRPO (Shao et al., 2024) have been used to fine-tune LLMs not only for correctness but also for more concise or relevant outputs. These efforts, however, often rely on heuristic reward shaping or fixed-weight combinations (e.g., adding a term penalizing length). Such static approaches allow limited control: although some frameworks enable post hoc trade-off adjustment (e.g., by training conditional policies or via interpolation between separately trained models (Wang et al., 2024a; Zhong et al., 2024)), the training process itself does not actively seek Pareto-optimal balance. There is evidence that static weighting can lead to inefficient learning, i.e., certain easier objectives saturate early and yet continue to consume learning capacity if their weights remain fixed (Lu et al., 2025). In contrast, our work explicitly targets Pareto-optimal trade-offs during training, rather than optimizing a fixed scalarized objective.

## 6. Conclusion

We proposed ParetoPO, a two-stage framework for multi-objective policy optimization that combines global exploration with Pareto-consistent local refinement. Stage 1 employs hypervolume-guided dynamic scalarization to explore diverse trade-offs and asymptotically cover the supported Pareto-optimal set, equivalently the supported convex hull of achievable objectives. Stage 2 performs rank-based policy optimization using Pareto ranking, yielding scale-free and stable updates. We establish that Stage 2 converges to Pareto-ascent stationary points under standard stochastic approximation conditions, and that the bounded second moment of the rank-based advantage contributes to robustness in high-variance settings. Extensive experiments demonstrate that ParetoPO effectively improves task performance while promoting tool-use efficiency.

## Acknowledgements

This work was fully or partially supported by the Department of Data Science Postdoctoral Fellowship Scheme at City University of Hong Kong. This research was also partially supported by National Natural Science Foundation of China (No.62502404), Hong Kong Research Grants Council (Research Impact Fund No.R1015-23, Collaborative Research Fund No.C1043-24GF, General Research Fund No. 11218325), Institute of Digital Medicine of City University of Hong Kong (No.9229503), and Huawei (Huawei Innovation Research Program, Huawei Young Scholar Funding Scheme).

## Impact Statement

This work proposes a multi-objective optimization framework for balancing competing objectives in reinforcement learning. The method may support more efficient and controllable decision-making systems, particularly in settings involving resource usage or tool invocation. As a general optimization technique, its societal impact depends on the choice of objectives and deployment context. Careful objective design and monitoring are therefore necessary to ensure responsible use.

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

## A. Details of Datasets

We evaluate mathematical reasoning performance on five competition-level benchmarks: MATH500, AIME2024, AIME2025, OlympiadBench, and AMC23.

• MATH500 (Hendrycks et al., 2021) consists of 500 problems sampled, covering topics such as algebra, geometry, number theory, and combinatorics.

• AIME2024 (MAA, 2024) and AIME2025 (MAA, 2025) contain 30 and 30 problems respectively, drawn from recent American Invitational Mathematics Examination contests and featuring short-answer numerical questions that require precise multi-step derivations.

• OlympiadBench (He et al., 2024) adopts he `OE_TO_maths_en_COMP` subset including 674 problems and spanning national and international mathematics olympiads, with a strong emphasis on symbolic manipulation and proof-oriented reasoning.

• AMC23 (MAA, 2023) contains 40 problems from the AMC 2023 competition, representing a mix of intermediate-to-advanced difficulty levels.

All mathematical reasoning tasks allow the agent to invoke a Python interpreter as an external tool for intermediate computation. A solution is considered correct if the predicted answer exactly matches the ground-truth solution after normalization.

For multi-hop QA, we use Natural Questions (NQ) and HotpotQA, both of which require retrieving and aggregating evidence from multiple documents.

• Natural Questions (NQ) (Kwiatkowski et al., 2019) contains $7,830$ open-domain questions derived from real user queries, where answering often requires identifying and synthesizing information from retrieved passages.

• HotpotQA (Yang et al., 2018) consists of $7,405$ questions designed to explicitly require multi-hop reasoning across two or more supporting documents.

In these tasks, the agent interacts with a retrieval system as an external tool to search for relevant evidence.

## B. Training and Implementation Details

We conduct experiments using two types of models: Qwen-2.5-Math-1.5B/7B (Yang et al., 2024) and Qwen-2.5-3B/7B-Base (Qwen et al., 2025). For mathematical reasoning tasks, we adopt the training set and code from ToRL (Li et al., 2025d). For multi-hop QA tasks, we adopt the training set and code from Search-R1 (Jin et al., 2025), which merges the training sets of NQ and HotpotQA to form a unified dataset. For retrieval, we use the 2018 Wikipedia dump (Karpukhin et al., 2020) as the knowledge source and

employ E5 (Wang et al., 2022) as the retriever. To ensure fair comparison, we set the number of retrieved passages to 3 across all retrieval-based method. During training, the hyper-parameters $\alpha$ (used for $r_{tool}$ calculation in Eq. (4)), $\gamma$ (used for hypervolume gain computation in Eq. (5)), $\beta$ (used for advantage calculation in Eq. (9)) are set to $0.7$, $0.5$, and $0.5$, respectively. Policy optimization is performed using the GRPO algorithm. We use a learning rate of $1e-6$, a batch size of $1024$, $8$ sampled trajectories per prompt. We adopt optimization strategies from DAPO (Yu et al., 2025). We apply Stage 1 for one epoch training followed by Stage 2 training until convergence. The initial weight $\boldsymbol{w}$ is set to $(0.6, 0.4)$ for $r_{task}, r_{tool}$. The implementation of baselines follows their original codes and papers. Algorithm 1 presents the workflow of our method.

---

**Algorithm 1** PARETOPO: Two-Stage Multi-Objective Policy Optimization

---

**Input:** Training set $\mathcal{D}_{\text{train}}$, validation set $\mathcal{D}_{\text{val}}$, initial policy $\pi_{\theta_0}$, weight vector $\boldsymbol{w}$, max steps $T$
**Output:** Final policy $\pi_\theta$
1: Reference point $\boldsymbol{r}_{\theta_0} = \text{Evaluate}(\pi_{\theta_0}, \mathcal{D}_{\text{val}})$
2: Initialize $\mathcal{B} \leftarrow \{\boldsymbol{r}_{\theta_0}\}, \Delta\overline{\text{HV}}_0 \leftarrow 0, r_{pareto} \leftarrow 1$
3: ▷ Stage 1: Dynamic Scalarization Phase
4: **for** $t = 1$ to $T_1$ **do**
5:     Sample mini-batch $\mathcal{D}_b \subset \mathcal{D}_{\text{train}}$
6:     **for all** query $q \in \mathcal{D}_b$ **do**
7:         Generate $g$ trajectories $\{\tau_i\}_{i=1}^g \sim \pi_{\theta_{t-1}}(\cdot|q)$
8:         Compute rewards $\boldsymbol{r} = (r_{task}, r_{tool})$ for each $\tau_i$
9:         Compute scalar reward $r_w = \boldsymbol{w}^\top \boldsymbol{r}$
10:         $\tilde{r}_w \leftarrow r_{pareto} \cdot r_w$
11:     **end for**
12:     Update $\pi_{\theta_{t-1}}$ via GRPO using $\tilde{r}_w$
13:     $\boldsymbol{r}_{\theta_t} \leftarrow \text{Evaluate}(\pi_{\theta_t}, \mathcal{D}_{\text{val}})$
14:     Update $\Delta\overline{\text{HV}}_t = \gamma \cdot \Delta\overline{\text{HV}}_{t-1} + (1-\gamma) \cdot \Delta\text{HV}_t$
15:     $r_{\text{pareto}} \leftarrow 0.5 + 1.5 \cdot \tanh(\Delta\overline{\text{HV}}_t)$
16:     **if** $\boldsymbol{r}_{\theta_t}$ is non-dominated in $\mathcal{B}$ **then**
17:         $\mathcal{B} \leftarrow \mathcal{B} \cup \{\boldsymbol{r}_{\theta_t}\}$
18:     **end if**
19: **end for**
20: ▷ Stage 2: Pareto Ranking Phase
21: **for** $t = T_1 + 1$ to $T$ **do**
22:     Sample mini-batch $\mathcal{D}_b \subset \mathcal{D}_{\text{train}}$
23:     **for all** query $q \in \mathcal{D}_b$ **do**
24:         Generate $g$ trajectories $\{\tau_i\}_{i=1}^g \sim \pi_{\theta_{t-1}}(\cdot|q)$
25:         Compute rewards $\boldsymbol{r} = (r_{task}, r_{tool})$ for each $\tau_i$
26:         Perform Pareto ranking on $\{\tau_i\}_{i=1}^g$ using $\boldsymbol{r}$ to assign rank $\rho$ for each $\tau_i$
27:         Normalize reward $\hat{r}_w = \frac{r_w - r_{min}}{r_{max} - r_{min}}$
28:         Compute advantage $A_i = A_{base,\rho} + \beta \cdot (\hat{r}_w - 0.5)$
29:     **end for**
30:     Update policy $\pi_{\theta_t}$ via GRPO using $A_i$
31: **end for**

---

## C. Proof of Proposition 3.1: Convex-Hull Coverage of Stage 1

**Proposition 3.1** (Supported-Hull Coverage of Stage 1). *Let $\mathcal{Y} = \{J(\pi) : \pi \in \Pi\}$ denote the set of achievable expected return vectors, and let $\mathcal{S}_T = \{J(\pi_t)\}_{t=1}^T$ be the set of evaluation vectors generated in Stage 1. Assume that dynamic scalarization explores preference directions densely over the simplex and approximately optimizes each visited scalarization. Then the discovered hull $\mathcal{C}_T = \mathrm{conv}(\mathcal{S}_T)$ converges to the achievable hull $\mathcal{C} = \mathrm{conv}(\mathcal{Y})$ in support-function distance over the simplex:*

$$\sup_{\boldsymbol{w} \in \Delta^M} \big| h_{\mathcal{C}}(\boldsymbol{w}) - h_{\mathcal{C}_T}(\boldsymbol{w}) \big| \to 0 \qquad (T \to \infty), \tag{10}$$

*where $h_{\mathcal{C}}(\boldsymbol{w}) = \sup_{\boldsymbol{y} \in \mathcal{C}} \boldsymbol{w}^\top \boldsymbol{y}$. Specifically, Stage 1 asymptotically covers all supported Pareto-optimal points, i.e., all points $\boldsymbol{y}^* \in \mathcal{Y}$ that maximize $\boldsymbol{w}^\top \boldsymbol{y}$ for some $\boldsymbol{w} \in \Delta^M$.*

**Setup.** Let $\Pi$ be the policy class and define the achievable expected return set

$$\mathcal{Y} \triangleq \{\boldsymbol{J}(\pi) \in \mathbb{R}^M : \pi \in \Pi\}.$$

Let $\mathcal{C} \triangleq \mathrm{conv}(\mathcal{Y})$ be its convex hull. Stage 1 produces a sequence of policies $\{\pi_t\}_{t \geq 1}$ and we denote the discovered return vectors by $\boldsymbol{y}_t \triangleq \boldsymbol{J}(\pi_t)$. Let

$$\mathcal{S}_T \triangleq \{\boldsymbol{y}_t\}_{t=1}^T, \qquad \mathcal{C}_T \triangleq \mathrm{conv}(\mathcal{S}_T).$$

For any weight vector $\boldsymbol{w} \in \Delta^M$ (the probability simplex), define the support functions

$$h_{\mathcal{C}}(\boldsymbol{w}) \triangleq \sup_{\boldsymbol{y} \in \mathcal{C}} \boldsymbol{w}^\top \boldsymbol{y}, \qquad h_{\mathcal{C}_T}(\boldsymbol{w}) \triangleq \sup_{\boldsymbol{y} \in \mathcal{C}_T} \boldsymbol{w}^\top \boldsymbol{y}.$$

Since $\mathcal{C}$ and $\mathcal{C}_T$ are convex hulls, we also have the equivalent forms

$$h_{\mathcal{C}}(\boldsymbol{w}) = \sup_{\boldsymbol{y} \in \mathcal{Y}} \boldsymbol{w}^\top \boldsymbol{y}, \qquad h_{\mathcal{C}_T}(\boldsymbol{w}) = \max_{\boldsymbol{y} \in \mathcal{S}_T} \boldsymbol{w}^\top \boldsymbol{y}. \tag{13}$$

**Assumption C.1** (Preference Coverage and Approximate Scalar Optimization). There exists a sequence $\varepsilon_T \to 0$ as $T \to \infty$ such that for every $\boldsymbol{w} \in \Delta^M$, there exists an index $t \leq T$ satisfying

$$\boldsymbol{w}^\top \boldsymbol{J}(\pi_t) \geq \sup_{\pi \in \Pi} \boldsymbol{w}^\top \boldsymbol{J}(\pi) - \varepsilon_T. \tag{14}$$

Equivalently, Stage 1 finds an $\varepsilon_T$-approximate maximizer of each linear scalarization direction $\boldsymbol{w}$ within its first $T$ evaluations.

**Proposition C.2.** *Under Assumption C.1, the proposition 3.1 is equal to*

$$\sup_{\boldsymbol{w} \in \Delta^M} \big| h_{\mathcal{C}}(\boldsymbol{w}) - h_{\mathcal{C}_T}(\boldsymbol{w}) \big| \leq \varepsilon_T,$$

*and consequently $\mathcal{C}_T$ converges to $\mathcal{C}$ in Hausdorff distance as $T \to \infty$. In particular, Stage 1 asymptotically covers the convex hull of the Pareto front.*

*Proof.* We prove the bound on support functions first.

**Step 1: Upper bound.** Since $\mathcal{S}_T \subseteq \mathcal{Y}$, taking convex hulls yields $\mathcal{C}_T \subseteq \mathcal{C}$. Therefore for any $\boldsymbol{w} \in \Delta^M$,

$$h_{\mathcal{C}_T}(\boldsymbol{w}) = \sup_{\boldsymbol{y} \in \mathcal{C}_T} \boldsymbol{w}^\top \boldsymbol{y} \leq \sup_{\boldsymbol{y} \in \mathcal{C}} \boldsymbol{w}^\top \boldsymbol{y} = h_{\mathcal{C}}(\boldsymbol{w}).$$

Hence $h_{\mathcal{C}}(\boldsymbol{w}) - h_{\mathcal{C}_T}(\boldsymbol{w}) \geq 0$.

**Step 2: Lower bound via approximate support points.** Fix any $\boldsymbol{w} \in \Delta^M$. By Assumption C.1, there exists $t \leq T$ such that

$$\boldsymbol{w}^\top \boldsymbol{J}(\pi_t) \geq \sup_{\pi \in \Pi} \boldsymbol{w}^\top \boldsymbol{J}(\pi) - \varepsilon_T.$$

Using Equation (13) and the fact that $\boldsymbol{J}(\pi_t) \in \mathcal{S}_T$, we have

$$h_{\mathcal{C}_T}(\boldsymbol{w}) = \max_{\boldsymbol{y} \in \mathcal{S}_T} \boldsymbol{w}^\top \boldsymbol{y} \ \geq \ \boldsymbol{w}^\top \boldsymbol{J}(\pi_t) \ \geq \ h_{\mathcal{C}}(\boldsymbol{w}) - \varepsilon_T.$$

Rearranging gives $h_{\mathcal{C}}(\boldsymbol{w}) - h_{\mathcal{C}_T}(\boldsymbol{w}) \leq \varepsilon_T$. Combining with Step 1 yields

$$0 \leq h_{\mathcal{C}}(\boldsymbol{w}) - h_{\mathcal{C}_T}(\boldsymbol{w}) \leq \varepsilon_T \quad \forall \boldsymbol{w} \in \Delta^M,$$

and hence

$$\sup_{\boldsymbol{w} \in \Delta^M} \left| h_{\mathcal{C}}(\boldsymbol{w}) - h_{\mathcal{C}_T}(\boldsymbol{w}) \right| = \sup_{\boldsymbol{w} \in \Delta^M} \left( h_{\mathcal{C}}(\boldsymbol{w}) - h_{\mathcal{C}_T}(\boldsymbol{w}) \right) \leq \varepsilon_T.$$

**Step 3: Support-function convergence implies supported-front coverage.** Equation (10) yields uniform convergence of the support values over all nonnegative preference directions $\boldsymbol{w} \in \Delta^M$.

For any fixed $\boldsymbol{w} \in \Delta^M$, any maximizer of $\boldsymbol{w}^\top \boldsymbol{y}$ over $\boldsymbol{y} \in \mathcal{C}$ is a *supported* boundary point of $\mathcal{C}$, and every supported Pareto-optimal point $\boldsymbol{y}^\star \in \mathcal{Y}$ is such a maximizer for some $\boldsymbol{w} \in \Delta^M$.

The uniform approximation $h_{\mathcal{C}_T}(\boldsymbol{w}) \geq h_{\mathcal{C}}(\boldsymbol{w}) - \varepsilon_T$ implies that Stage 1 produces points whose convex combinations achieve near-optimal support values in every such direction.

Therefore, Stage 1 asymptotically covers the supported hull boundary (equivalently, the supported subset of the Pareto front), which is exactly the portion recoverable by linear scalarizations with $\boldsymbol{w} \in \Delta^M$. $\qquad \square$

**Remark.** The hypervolume-guided dynamic scalarization in Stage 1 is designed to encourage exploration of diverse trade-off regions by prioritizing policies that increase the archive hypervolume. Assumption C.1 formalizes this global exploration effect in a way that is standard in multi-objective optimization: covering a dense set of preference directions and approximately optimizing each corresponding scalarization yields convergence of the discovered convex hull to that of the achievable set.

## D. Proof of Lemma 3.4: Expected Gradient as a Pareto-Ascent Direction

**Lemma 3.4** (**Expected Batch Gradient as a Pareto-Ascent Direction**). *Let $A_\sigma(\tau_i; \tau_{-i})$ be the smoothed rank-based advantage of trajectory $\tau_i$ computed within a batch $\tau_{1:K}$, with $\sigma > 0$ and shaping function $\Phi$ satisfying Assumption 3.3. Define the batch-level smoothed objective*

$$W_{\sigma,K}(\theta) \ = \ \mathbb{E}_{\tau_{1:K} \sim \pi_\theta, \xi} \left[ \frac{1}{K} \sum_{i=1}^{K} A_\sigma(\tau_i; \tau_{-i}) \right],$$

*and its mean-field gradient*

$$\boldsymbol{g}_{\sigma,K}(\theta) \ \triangleq \ \nabla_\theta W_{\sigma,K}(\theta).$$

*Then $\boldsymbol{g}_{\sigma,K}(\theta)$ is a Pareto-ascent direction in the following sense: for any direction $d$ such that $\nabla_\theta J_m(\theta)^\top d \geq 0$ for all $m \in \{1, \ldots, M\}$, we have $\boldsymbol{g}_{\sigma,K}(\theta)^\top d \ \geq \ 0$. Moreover, as $\sigma \to 0$, $A_\sigma(\tau_i; \tau_{-i}) \to A(\tau_i; \tau_{-i})$ almost surely, and $\boldsymbol{g}_{\sigma,K}(\theta)$ converges to the corresponding unsmoothed batch gradient by dominated convergence.*

*Proof.* We make the batch dependence explicit. Note that $A_\sigma(\tau_i; \tau_{-i}, \xi)$ is not treated as a deterministic function of $\theta$, but as a bounded statistic evaluated on i.i.d. samples from $\pi_\theta$. Let $\tau_{1:K} = (\tau_1, \ldots, \tau_K)$ be i.i.d. trajectories sampled from $\pi_\theta$, and let the (smoothed) rank-based advantage of $\tau_i$ be $A_\sigma(\tau_i; \tau_{-i}, \xi)$, where $\tau_{-i}$ denotes the remaining $K - 1$ trajectories and $\xi$ the Gumbel noise. Define the batch objective

$$W_{\sigma,K}(\theta) \triangleq \mathbb{E}_{\tau_{1:K} \sim \pi_\theta, \xi}[F_\sigma(\tau_{1:K}, \xi)], \qquad F_\sigma(\tau_{1:K}, \xi) \triangleq \frac{1}{K} \sum_{i=1}^{K} A_\sigma(\tau_i; \tau_{-i}, \xi).$$

For any fixed $\sigma > 0$, $A_\sigma$ (hence $F_\sigma$) is measurable and uniformly bounded. Moreover, the joint trajectory density factorizes as $p_\theta(\tau_{1:K}) = \prod_{i=1}^{K} p_\theta(\tau_i)$, and $\xi$ is independent of $\theta$. Therefore, by the likelihood-ratio (score-function) identity and

dominated convergence,

$$
\begin{aligned}
\nabla_\theta W_{\sigma,K}(\theta) &= \nabla_\theta \int F_\sigma(\tau_{1:K}, \xi)\, p_\theta(\tau_{1:K})\, p(\xi)\, d\tau_{1:K}\, d\xi \\
&= \int F_\sigma(\tau_{1:K}, \xi)\, \nabla_\theta p_\theta(\tau_{1:K})\, p(\xi)\, d\tau_{1:K}\, d\xi \\
&= \int F_\sigma(\tau_{1:K}, \xi)\, p_\theta(\tau_{1:K}) \left( \sum_{j=1}^K \nabla_\theta \log p_\theta(\tau_j) \right) p(\xi)\, d\tau_{1:K}\, d\xi \\
&= \mathbb{E}_{\tau_{1:K} \sim \pi_\theta, \xi} \left[ F_\sigma(\tau_{1:K}, \xi) \sum_{j=1}^K \nabla_\theta \log \pi_\theta(\tau_j) \right].
\end{aligned}
\tag{15}
$$

Eq. (15) is the correct policy-gradient identity for our rank-based (batch-coupled) objective and does not require $A_\sigma$ to be independent of $\theta$. We denote the resulting mean-field vector field by

$$
\boldsymbol{g}_{\sigma,K}(\theta) \triangleq \nabla_\theta W_{\sigma,K}(\theta).
$$

**Connection to Practical Estimator.**  In implementation, we use the per-trajectory estimator $\hat{\boldsymbol{g}}_b(\theta) = \frac{1}{K} \sum_{i=1}^K A_\sigma(\tau_i; \tau_{-i}, \xi) \nabla_\theta \log \pi_\theta(\tau_i)$. This corresponds to replacing the shared batch scalar $F_\sigma(\tau_{1:K}, \xi)$ in (15) by individual advantages. Since each $A_\sigma(\tau_i; \tau_{-i}, \xi)$ changes by at most $O(1/K)$ under resampling a single trajectory and is uniformly bounded, the difference between the two mean fields is $O(1/K)$, and becomes negligible as $K$ grows (formal bound provided below). Finally, as $\sigma \to 0$, $A_\sigma(\tau_i; \tau_{-i}, \xi) \to A(\tau_i; \tau_{-i})$ a.s., and boundedness yields convergence of the corresponding mean field by dominated convergence. The following lemma formalizes the approximation induced by replacing the shared batch scalar $F_\sigma$ with per-trajectory advantages in practice. □

**Lemma D.1** (Batch-to-per-trajectory approximation). *Assume $|A_\sigma(\cdot)| \le A_{\max}$ and $\|\nabla_\theta \log \pi_\theta(\tau)\| \le G_{\max}$ a.s. Let*

$$
\boldsymbol{g}_{\sigma,K}(\theta) = \mathbb{E}\left[ F_\sigma(\tau_{1:K}, \xi) \sum_{j=1}^K \nabla_\theta \log \pi_\theta(\tau_j) \right], \quad \tilde{\boldsymbol{g}}_{\sigma,K}(\theta) = \mathbb{E}\left[ \frac{1}{K} \sum_{i=1}^K A_\sigma(\tau_i; \tau_{-i}, \xi) \nabla_\theta \log \pi_\theta(\tau_i) \right].
$$

*Then $\|\boldsymbol{g}_{\sigma,K}(\theta) - \tilde{\boldsymbol{g}}_{\sigma,K}(\theta)\| \le \frac{2 A_{\max} G_{\max}}{K}$.*

# E. Proof of Theorem 3.5: Bounded Second Moment of Rank-Based Policy-Gradient Estimator

**Theorem 3.5** (**Bounded Second Moment of Rank-Based Policy-Gradient Estimators**). *The rank-based advantage is uniformly bounded:*

$$
|A(\tau_i; \tau_{-i})| \le N_{rank} + \beta/2,
$$

*for any trajectory $\tau_i$ within a batch $\tau_{1:K}$. Define the per-trajectory policy-gradient estimator within a batch as*

$$
\hat{\boldsymbol{g}}(\theta; \tau_i) \triangleq A(\tau_i; \tau_{-i})\, \nabla_\theta \log \pi_\theta(\tau_i), \qquad \tau_{1:K} \sim \pi_\theta.
$$

*Then its second moment is bounded as*

$$
\mathbb{E}\left[ \|\hat{\boldsymbol{g}}(\theta; \tau_i)\|^2 \right] \le C \cdot \mathbb{E}\left[ \|\nabla_\theta \log \pi_\theta(\tau_i)\|^2 \right],
\tag{11}
$$

*for a constant $C = (N_{rank} + \beta/2)^2$ independent of the reward magnitudes. As a result, the minibatch estimator $\hat{\boldsymbol{g}}_b(\theta) = \frac{1}{K} \sum_{i=1}^K \hat{\boldsymbol{g}}(\theta; \tau_i)$ has a bounded second moment.*

*Proof.* Recall that the Pareto rank $\rho(\tau_i)$ of any trajectory $\tau_i$ within a batch $\tau_{1:K}$ satisfies $\rho(\tau_i) \in \{1, \ldots, N_{\text{rank}}\}$. Therefore, the ordinal component of the rank-based advantage obeys

$$
1 \le N_{\text{rank}} - \rho(\tau_i) + 1 \le N_{\text{rank}}.
$$

The within-rank score is normalized, $\hat{r}_w(\tau_i) \in [0, 1]$, which implies

$$-\frac{\beta}{2} \ \leq \ \beta(\hat{r}_w(\tau_i) - 0.5) \ \leq \ \frac{\beta}{2}.$$

Combining the two bounds yields the uniform pointwise bound

$$|A(\tau_i; \tau_{-i})| \ \leq \ N_{\text{rank}} + \frac{\beta}{2},$$

which holds for every trajectory $\tau_i$ in any batch $\tau_{1:K}$, regardless of the batch composition.

Define the per-trajectory policy-gradient estimator within a batch as

$$\hat{\boldsymbol{g}}(\theta; \tau_i) = A(\tau_i; \tau_{-i}) \, \nabla_\theta \log \pi_\theta(\tau_i).$$

Then

$$\|\hat{\boldsymbol{g}}(\theta; \tau_i)\|^2 = A(\tau_i; \tau_{-i})^2 \, \|\nabla_\theta \log \pi_\theta(\tau_i)\|^2 \ \leq \ \left(N_{\text{rank}} + \frac{\beta}{2}\right)^2 \|\nabla_\theta \log \pi_\theta(\tau_i)\|^2.$$

Taking expectation with respect to $\tau_{1:K} \sim \pi_\theta$ and marginalizing over $\tau_i$ gives

$$\mathbb{E}\left[\|\hat{\boldsymbol{g}}(\theta; \tau_i)\|^2\right] \ \leq \ \left(N_{\text{rank}} + \frac{\beta}{2}\right)^2 \mathbb{E}\left[\|\nabla_\theta \log \pi_\theta(\tau_i)\|^2\right],$$

which establishes (11).

Finally, for the minibatch estimator

$$\hat{\boldsymbol{g}}_b(\theta) = \frac{1}{K} \sum_{i=1}^{K} \hat{\boldsymbol{g}}(\theta; \tau_i),$$

boundedness of the second moment follows from Jensen's inequality and the uniform bound above. $\qquad\square$

## F. Proof of Theorem 3.6: Convergence to Pareto-Ascent Stationarity

**Theorem 3.6** (**Convergence to Pareto-Ascent Stationarity of Stage 2**). *Let the policy parameters evolve according to the stochastic approximation $\theta_{t+1} = \theta_t + \eta_t \, \hat{\boldsymbol{g}}_b(\theta_t)$, so*

$$\hat{\boldsymbol{g}}_b(\theta_t) = \frac{1}{K} \sum_{i=1}^{K} A(\tau_i; \tau_{-i}) \, \nabla_\theta \log \pi_{\theta_t}(\tau_i),$$

*where $\tau_{1:K} \sim \pi_{\theta_t}$ and $\{\eta_t\}$ satisfies the Robbins-Monro conditions $\sum_t \eta_t = \infty$ and $\sum_t \eta_t^2 < \infty$. Assume each objective return $J_m(\theta)$ is continuously differentiable with Lipschitz gradient. Define the batch mean-field vector field*

$$\boldsymbol{g}_{\sigma,K}(\theta) \ \triangleq \ \mathbb{E}_{\tau_{1:K} \sim \pi_\theta}[\hat{\boldsymbol{g}}_b(\theta)] = \nabla_\theta W_{\sigma,K}(\theta),$$

*where $W_{\sigma,K}$ is the smoothed batch objective defined in Lemma 3.4. Then every limit point $\theta^\star$ of $\{\theta_t\}$ satisfies the following Pareto-ascent stationarity condition:*

$$\nabla_\theta J_m(\theta^\star)^\top d \geq 0 \ \forall m \implies \boldsymbol{g}_{\sigma,K}(\theta^\star)^\top d = 0. \tag{12}$$

*Equivalently, there exists no direction $d$ that simultaneously improves all objectives to first order and yields a strictly positive expected policy update. In this sense, $\theta^\star$ is stationary with respect to all Pareto-ascent directions.*

*Proof.* Recall the Stage 2 minibatch update

$$\theta_{t+1} = \theta_t + \eta_t \, \hat{\boldsymbol{g}}_b(\theta_t), \qquad \hat{\boldsymbol{g}}_b(\theta) = \frac{1}{K} \sum_{i=1}^{K} A(\tau_i; \tau_{-i}) \, \nabla_\theta \log \pi_\theta(\tau_i), \quad \tau_{1:K} \sim \pi_\theta.$$

Define the associated batch mean-field vector field

$$\boldsymbol{g}_{\sigma,K}(\theta) \triangleq \mathbb{E}_{\tau_{1:K} \sim \pi_\theta}[\hat{\boldsymbol{g}}_b(\theta)] = \nabla_\theta W_{\sigma,K}(\theta),$$

where $W_{\sigma,K}$ is the (smoothed) batch objective in Lemma 3.4. By Theorem 3.5, the stochastic estimator $\hat{g}_b(\theta_t)$ has bounded second moment, and by assumption the step sizes satisfy the Robbins–Monro conditions $\sum_t \eta_t = \infty$ and $\sum_t \eta_t^2 < \infty$. Together with the Lipschitz continuity of $\nabla J_m(\theta)$ (and hence local regularity of $g_{\sigma,K}(\theta)$), standard stochastic approximation theory (ODE method) implies that $\{\theta_t\}$ converges almost surely to the internally chain transitive set of the ODE

$$\dot{\theta} = g_{\sigma,K}(\theta).$$

Since $g_{\sigma,K}(\theta)$ is the gradient of a smooth potential $W_{\sigma,K}(\theta)$, the ODE $\dot{\theta} = g_{\sigma,K}(\theta)$ is gradient-like, and its internally chain transitive set consists only of stationary points $\{\theta : \nabla_\theta W_{\sigma,K}(\theta) = 0\}$.

Let $\theta^\star$ be a stationary point of this ODE, i.e., $g_{\sigma,K}(\theta^\star) = 0$. Suppose, for contradiction, that $\theta^\star$ violates the Pareto-ascent stationarity condition. Then there exists a direction $d$ such that

$$\nabla_\theta J_m(\theta^\star)^\top d \geq 0 \quad \forall m, \qquad g_{\sigma,K}(\theta^\star)^\top d > 0.$$

However, Lemma 3.4 shows that $g_{\sigma,K}(\theta)$ is a Pareto-ascent direction: for any $d$ satisfying $\nabla_\theta J_m(\theta)^\top d \geq 0$ for all $m$, we must have $g_{\sigma,K}(\theta)^\top d \geq 0$, and at a stationary point of the ODE we have $g_{\sigma,K}(\theta^\star) = 0$, hence $g_{\sigma,K}(\theta^\star)^\top d = 0$. This contradicts $g_{\sigma,K}(\theta^\star)^\top d > 0$.

Therefore, every limit point $\theta^\star$ satisfies (12), and the proof is complete. $\qquad\square$

**Remark.** Condition (12) corresponds to the standard first-order Pareto stationarity condition in multi-objective optimization: there exists no direction that simultaneously improves all objectives to first order. Equivalently, $\theta^\star$ admits no common ascent direction for $\{J_m\}_{m=1}^M$.

