# OpenReview forum: "Towards Pareto-Optimal Tool-Integrated Agents with Pareto Ranking Policy Optimization"
_ICML.cc/2026/Conference — ICML 2026 spotlight_

### Official Review · Reviewer_QmdP · 2026-03-12

**Soundness:** 3
**Presentation:** 3
**Significance:** 3
**Originality:** 3
**Overall Recommendation:** 4
**Confidence:** 3

**Summary:**

This paper studies the problem of multi-objective alignment for tool-integrated LLM agents. It is about how to train LLM agents under multiple conflicting objectives using multi-objective reinforcement learning.

**Compliance With Llm Reviewing Policy:**

Affirmed.

**Final Justification:**

This is a sound and clearly presented work. The authors’ rebuttal has addressed my concerns, and I maintain a weak accept recommendation while increasing my confidence. I thank the authors for their efforts and their rebuttal.

**Key Questions For Authors:**

1. How does the Pareto ranking mechanism scale as the number of objectives increases? Does the number of non-dominated solutions typically explode?
2. Have the authors tested if the efficiency gains found on one dataset (e.g., MATH500) generalize to unseen tasks without further fine-tuning?

**Limitations:**

See weakness and questions.

**Strengths And Weaknesses:**

Strength
1. The motivation is interesting and the problem is important. Agents should balance multiple factors such as accuracy, cost, latency, and tool usage in practice.
2. The authors formulate the agent training problem as a multi-objective Markov Decision Process with two reward components: task success and tool efficiency. The authors provide solid theoretical analysis to strengthen the methodological rigor of the work.

Weakness
1. The experiments only consider two objectives. It would be helpful to evaluate the method with more objectives to demonstrate its scalability and generality.
2. All experiments appear to use relatively small models (e.g., Qwen 1.5B or 7B). It is unclear whether the method scales effectively to larger LLMs. In fact, LLMs with larger parameter sizes require a more careful trade-off between performance and efficiency.
3. The introduction of hypervolume contribution calculations and non-dominated sorting introduces additional computational overhead. The paper does not provide a detailed analysis of the training overhead compared to baselines.

---

> ### Author Rebuttal · Authors · 2026-03-31
>
> Thank you for your insightful comments.
>
> >**W1: Evaluation on only two objectives.**
>
> To further verify that our method can extend beyond two objectives, we follow the setup in previous work [1] and conduct an additional three-objective experiment with Accuracy, Response Length (number of tokens), and Clarity (clear reasoning process). As shown in the table below, our method still outperforms the previous baselines in this setting. Besides, the theoretical analysis in Section 3.3 does not limit the number of objectives and further proves the feasibility of extending our method to more objectives from a theoretical view.
>
> |          |  |        MATH500 (Qwen2.5-Math-1.5B)         |         | |        NQ (Qwen2.5-3B-Base)          |         |
> |----------|-----------------------------|-----------------|---------|----------------------|-----------------|---------|
> |          | Accuracy                    | Response_length | Clarity | Accuracy             | Response_length | Clarity |
> | OTC-GRPO | 74.0                        | 1778            | 0.945   | 44.4                 | 1778            | 0.932   |
> | MO-GRPO  | 71.2                        | 1675            | 0.953   | 41.2                 | 1478            | 0.946   |
> | ParetoPO | 80.0                        | 1367            | 0.977   | 48.0                 | 1367            | 0.971   |
>
> [1] Lu, Y., Wang, Z., Li, S., Liu, X., Yu, C., Yin, Q., ... & Jiang, M. (2025). Learning to optimize multi-objective alignment through dynamic reward weighting. arXiv preprint arXiv:2509.11452.
>
> >**W2: All experiments appear to use relatively small models. It is unclear about the scaling to larger models.**
>
> We add the results of Qwen2.5-14B as below. We can see that our method can still outperform baselines.
>
> |          | MATH500 |            | NQ   |            |
> |----------|---------|------------|------|------------|
> |          | EM      | #tool_call | EM   | #tool_call |
> | OTC-GRPO | 85.6    | 1.3        | 50.1 | 1.0        |
> | MO-GRPO  | 83.4    | 1.7        | 45.7 | 1.3        |
> | ParetoPO | 89.0    | 1.1        | 57.7 | 0.8        |
>
> >**W3: Analysis of the training overhead for hypervolume calculations and non-dominated sorting.**
>
> To better assess the implementation overhead, we randomly sampled rewards under different numbers of objectives and measured the average runtime of hypervolume computation and Pareto-based ranking on 1,000 samples. As shown in the table below, the computational cost increases with the number of objectives, but does not exhibit exponential growth.
>
> (ms)
> | #Objectives             | 2      | 4      | 6      | 8      | 10     |
> |-------------------------|--------|--------|--------|--------|--------|
> | Hypervolume computation | 5.43   | 10.41  | 10.42  | 19.03  | 25.23  |
> | Pareto-based  ranking          | 766.61 | 771.34 | 801.23 | 945.31 | 967.11 |
>
> To show the training overhead, we present the average runtime of prior baselines and our method variants（consistent with Ablation Study）：
> - Stage 1 (w/o dynamic weighting): Replacing dynamic scalarization with fixed-weight scalarization
> - Stage 2 (w/o Pareto Ranking): disabling Pareto ranking advantage estimation and using GRPO-style advantage estimation
>
> We can see that the training time for these two stages is slightly longer than that for the baseline, but it is still within a manageable range.
>
> |                                 | Runtime per step (seconds) |
> |---------------------------------|---------------------|
> | OTC-GRPO                        | 215.51              |
> | Stage 1                         | 226.31              |
> | Stage 1 (w/o dynamic weighting) | 217.82              |
> | Stage 2                         | 230.11              |
> | Stage 2 (w/o Pareto Ranking)    | 215.10              |
>
>
> >**Q1: How does the Pareto ranking mechanism scale as the number of objectives increases? Does the number of non-dominated solutions typically explode?**
>
> The approxiate runtime of Pareto ranking can refer to the W3 response. The computational cost increases with the number of objectives, but does not exhibit exponential growth. In our implementation, Pareto ranking is performed only within 8 sampled rollouts for each query. Therefore, even if the proportion of non-dominated trajectories increases with more objectives, this effect is bounded within a very small candidate set, so it does not lead to an uncontrolled explosion in ranking cost.
>
> >**Q2: Generalization to unseen tasks.**
>
> We have not evaluated cross-task transfer without further fine-tuning. We agree this is an important direction, but note that it is less straightforward in the tool-use setting, where different tasks often involve different tools and interfaces. In such cases, transfer performance would reflect not only method generalization, but also tool-space and task-structure mismatch. We will clarify this scope in the revision.

---

> > ### Author Rebuttal · Reviewer_QmdP · 2026-04-02
> >
> > Thank the authors for their response. I have no further questions or concerns. I will maintain my original score.

---

### Official Review · Reviewer_i3Wo · 2026-03-12

**Soundness:** 3
**Presentation:** 3
**Significance:** 3
**Originality:** 3
**Overall Recommendation:** 4
**Confidence:** 3

**Summary:**

The paper proposes ParetoPO, a two-stage multi-objective RL framework for tool-integrated LLM agents. It combines hypervolume-guided dynamic scalarization with Pareto-ranking advantages. Experiments show higher accuracy with fewer tool calls across math reasoning and multi-hop QA benchmarks.

**Compliance With Llm Reviewing Policy:**

Affirmed.

**Key Questions For Authors:**

- Possible data issue: identical AIME2024/2025 results for the 1.5B model raise concerns about whether distinct datasets were used.

**Limitations:**

yes

**Strengths And Weaknesses:**

Strength：
- Solid theoretical analysis with convergence guarantees.
- Consistent empirical improvements across math benchmarks and model sizes.
- Tool usage is significantly reduced while accuracy improves.

Weakness：
- The efficiency reward depends on a moving
$𝑁_{optimal}$, which may introduce training instability. Its dynamics and compatibility with the convergence analysis are not discussed.
- Experiments use only Qwen2.5 models; generalization to other backbones is unclear.

---

> ### Author Rebuttal · Authors · 2026-03-31
>
> Thank you for your valuable comments.
>
> >**W1: The efficiency reward depends on a moving N_optimal, which may introduce training instability. Its dynamics and compatibility with the convergence analysis are not discussed.**
>
> We agree that the moving $N_{optimal}$ introduces some non-stationarity, but we mitigate this in two ways.
>
> 1. First, $N_{optimal}$ is defined as the minimum tool count among successful trajectories, so it is monotonically non-increasing over training. Unlike a noisy reward baseline, it changes only when a more efficient successful trajectory is found, which makes its dynamics stable. In addition, the efficiency reward $r_{tool}$ is bounded in [0,1], preventing unstable reward magnitudes.
>
> 2. Second, as training progresses and the policy improves, discovering trajectories with even fewer tool calls becomes increasingly rare. As a result, $N_{optimal}$ quickly stabilizes and the optimization becomes approximately stationary in later stages. As shown in the table below, we observe that most updates to $N_{optimal}$ occur in the early phase of training, after which it remains largely stable.
>
> | training_step | 100 | 500 | 1000 | 1500 | 2000 |
> |---------------|-----|-----|------|------|------|
> | $N_{optimal}$   | 2.1 | 1.5 | 1.2  | 0.8  | 0.8  |
> | #tool_call    | 2.6 | 1.8 | 1.3  | 0.9  | 0.9  |
>
> >**W2: Experiments use only Qwen2.5 models; generalization to other backbones is unclear.**
>
> To allevitae the limitation in generalization of our method, we add the results of Llama-3.1-8B-Instruct as below. We can see that our method can still outperform previous baselines.
>
> |          | MATH500 |            | NQ   |            |
> |----------|---------|------------|------|------------|
> |          | EM      | #tool_call | EM   | #tool_call |
> | OTC-GRPO | 74.8    | 1.5        | 40.1 | 1.2        |
> | MO-GRPO  | 72.1    | 2.0        | 37.7 | 1.8        |
> | ParetoPO | 76.8    | 1.3        | 46.1 | 1.1        |
>
> >**Q1: Possible data issue: identical AIME2024/2025 results for the 1.5B model raise concerns about whether distinct datasets were used.**
>
> We checked both the evaluation code and the data, and found no issue. The 1.5B model also shows identical AIME2024/2025 results in prior work such as ToRL and OTC. A likely reason is that both benchmark test sets contain only 30 examples, making identical scores possible due to the coarse evaluation granularity.

---

> > ### Author Rebuttal · Reviewer_i3Wo · 2026-04-06
> >
> > Thank you for your careful and detailed response. I will keep my current score.

---

### Official Review · Reviewer_y1Pd · 2026-03-13

**Soundness:** 3
**Presentation:** 3
**Significance:** 3
**Originality:** 3
**Overall Recommendation:** 5
**Confidence:** 3

**Summary:**

This paper studies tool-integrated LLM agents from a multi-objective reinforcement learning perspective. It proposes ParetoPO, a two-stage method that first uses hypervolume-guided dynamic scalarization to explore better accuracy efficiency trade-offs, and then applies Pareto rank-based policy optimization to favor non-dominated trajectories during RL training. The experiments on MATH and QA tasks show that ParetoPO consistently improves the accuracy while reducing tool calls simultaneously.

**Compliance With Llm Reviewing Policy:**

Affirmed.

**Key Questions For Authors:**

N/A

**Limitations:**

yes

**Strengths And Weaknesses:**

Strength:
* The paper studies an important and practically meaningful problem: tool-integrated agents should optimize both task performance and tool-use efficiency. The proposed two-stage framework is novel and well motivated, and offers an inspiring way to cast tool-agent training as Pareto-aware multi-objective RL.
* The empirical results are strong. For both Math and QA benchmarks, ParetoPO consistently achieves the best overall trade-off between exact-match accuracy and tool call efficiency, and in many settings it improves accuracy while simultaneously reducing tool usage relative to prior GRPO-based baselines.
* The method appears relatively simple and practical to implement. It introduces only a small number of additional hyperparameters beyond standard GRPO training, and the sensitivity analysis suggests that the method is reasonably robust to different reward-weight choices on both math and QA tasks.
* The paper provides theoretical justification for both stages of the method, including coverage of supported Pareto-optimal points in Stage 1 and Pareto-ascent properties of the Stage 2 optimization procedure.

Weakness:
* The main weakness is scalability beyond the two-objective setup. The method is demonstrated only for task accuracy and tool efficiency, and it is unclear whether the proposed two-stage procedure would remain practical or effective when extended to three or more objectives. Since both hypervolume computation and Pareto-based ranking typically become more cumbersome in higher-dimensional objective spaces, the implementation and optimization complexity may grow quickly, limiting the generality of the approach.

---

> ### Author Rebuttal · Authors · 2026-03-31
>
> Thank you for your insightful comments.
>
> >**W1-1: The main weakness is scalability beyond the two-objective setup. The method is demonstrated only for task accuracy and tool efficiency, and it is unclear whether the proposed two-stage procedure would remain practical or effective when extended to three or more objectives.**
>
> To further verify that our method can extend beyond two objectives, we follow the setup in previous work [1] and conduct an additional three-objective experiment with Accuracy, Response Length (number of tokens), and Clarity (clear reasoning process). As shown in the table below, our method still outperforms the previous baselines in this setting. Besides, the theoretical analysis in Section 3.3 does not limit the number of objectives and further proves the feasibility of extending our method to more objectives from a theoretical view.
>
> |          |  |        MATH500 (Qwen2.5-Math-1.5B)         |         | |        NQ (Qwen2.5-3B-Base)          |         |
> |----------|-----------------------------|-----------------|---------|----------------------|-----------------|---------|
> |          | Accuracy                    | Response_length | Clarity | Accuracy             | Response_length | Clarity |
> | OTC-GRPO | 74.0                        | 1778            | 0.945   | 44.4                 | 1778            | 0.932   |
> | MO-GRPO  | 71.2                        | 1675            | 0.953   | 41.2                 | 1478            | 0.946   |
> | ParetoPO | 80.0                        | 1367            | 0.977   | 48.0                 | 1367            | 0.971   |
>
> [1] Lu, Y., Wang, Z., Li, S., Liu, X., Yu, C., Yin, Q., ... & Jiang, M. (2025). Learning to optimize multi-objective alignment through dynamic reward weighting. arXiv preprint arXiv:2509.11452.
>
> >**W1-2: Since both hypervolume computation and Pareto-based ranking typically become more cumbersome in higher-dimensional objective spaces, the implementation and optimization complexity may grow quickly, limiting the generality of the approach.**
>
> To better assess the implementation overhead, we randomly sampled rewards under different numbers of objectives and measured the average runtime (ms) of hypervolume computation and Pareto-based ranking on 1,000 samples. As shown in the table below, the computational cost increases with the number of objectives, but does not exhibit exponential growth. This suggests that the approach remains practical in low-dimensional multi-objective settings, while more scalable approximations or hypervolume surrogates are a promising direction for higher-dimensional cases.
>
> (ms)
> | #Objectives             | 2      | 4      | 6      | 8      | 10     |
> |-------------------------|--------|--------|--------|--------|--------|
> | Hypervolume computation | 5.43   | 10.41  | 10.42  | 19.03  | 25.23  |
> | Pareto-based   ranking         | 766.61 | 771.34 | 801.23 | 945.31 | 967.11 |
>
> As for the optimization complexity, we agree that as the number of objectives increases, the proportion of non-dominated solutions typically grows, which can reduce the discriminability of Pareto ranking and enlarge the frontier size. However, we view our method as general at the formulation level, while currently most practical for a small number of objectives (e.g., 2–3). The optimization on more objectives (e.g., 10) remains an important direction in the future. We will clarify this limitation more explicitly in the revision.

---

### Official Review · Reviewer_whi1 · 2026-03-16

**Soundness:** 2
**Presentation:** 3
**Significance:** 3
**Originality:** 3
**Overall Recommendation:** 4
**Confidence:** 4

**Summary:**

This submission introduces ParetoPO, a two-stage reinforcement learning framework tailored to optimize tool-augmented large language models across conflicting objectives, specifically task accuracy and tool-use efficiency. The first stage utilizes hypervolume-guided dynamic scalarization to adapt reward weights based on smoothed Pareto frontier progress, facilitating exploration of the supported convex hull. The second stage applies a Pareto-ranking-based policy optimization strategy, relying on non-dominated sorting to compute trajectory advantages within a Group Relative Policy Optimization (GRPO) setup. Empirical evaluations on mathematical reasoning benchmarks and multi-hop QA benchmarks demonstrate that ParetoPO produces policies with higher Exact Match scores and fewer tool calls than baseline methods.

**Compliance With Llm Reviewing Policy:**

Affirmed.

**Final Justification:**

My concerns have been adequately addressed, so updated my score accordingly.

**Key Questions For Authors:**

- In Section 2.3, $N_{optimal}$ is approximated using the minimum tool calls from current successful trajectories. How sensitive is the initial phase of Stage 1 to the stochastic discovery of highly sub-optimal success paths, and does this risk trapping the policy in a local efficiency minimum?
- In Table 1 and 2, it seems your optimization approach achieves both higher accuracy + higher efficiency than baselines, which is quite interesting. Considering the usual trade-off between performance vs. efficiency, most prior approaches focus on accuracy improvement (like Search-R1), what's the insight behind the strong empirical results that your multi-objective optimization approach can even outperform single-objective optimization approach?
- Experiment setting related:
1. For tool-integrated agents, it would be better to include more tool-use / agentic benchmarks like Tau-Bench (https://arxiv.org/abs/2406.12045) and BrowseComp (https://arxiv.org/abs/2504.12516).
2. Qwen-2.5 series are known to suffer from Spurious Rewards (https://arxiv.org/pdf/2506.10947), can you show the results with a different LLM family to support the generalizability of the proposed optimization method?
- Efficiency analysis of the proposed optimization method compared to baselines.

**Limitations:**

No, I do not see a "Limitations" section or paragraph discussing the limitations of the method. There is an Impact Statement though.

**Strengths And Weaknesses:**

Strengths:
- Multi-objective optimization is a critical and practical problem in aligning the behavior of LLM agents.
- The theoretical foundation explicitly connects the two-stage training approach to established multi-objective optimization principles.
- Integrating hypervolume-guided dynamic scalarization with discrete, non-dominated sorting for advantage computation within an online RL pipeline is a novel combination of techniques

Weaknesses:
- The experimental setting is not quite persuasive. From Table 1 / 2, it seems # tool calling is in general very small in the benchmarks selected (see Key Questions for more details).
- The clarity of the presentation can be further improved.
- As the experiments only focus on 2 objectives: performance vs. tool efficiency, the authors should scope down the claim of multi-objective optimization to be more specific, as it's not clear if the approach would generalize to 3+ objectives.

---

> ### Author Rebuttal · Authors · 2026-03-31
>
> Thank you for your reviews.
>
> >**W1: #Tool_call in Table 1 / 2 is very small.**
>
> For a fair comparison with prior multi-objective and tool-calling baselines (OTC-GRPO and ToRL), we follow their setups and use the same datasets. These datasets are relatively simple, leading to generally low tool-call counts. Therefore, we add results on BrowseComp (see the Q3 response).
>
> >**W2: The clarity of the presentation.**
>
> We will improve it in the next revision.
>
> >**W3: Evaluation on two objectives.**
>
> To verify the generalization of our method to more objectives, we follow the setup in prior work [1] and conduct a three-objective experiment with Accuracy, Response Length (number of tokens), and Clarity (clear reasoning process). As shown in the table below, our method still outperforms the baselines. Besides, the theoretical analysis in Section 3.3 does not limit the number of objectives and further proves the feasibility of extending our method to more objectives.
>
> |          |  |        MATH500 (Qwen2.5-Math-1.5B)         |         | |        NQ (Qwen2.5-3B-Base)          |         |
> |----------|-----------------------------|-----------------|---------|----------------------|-----------------|---------|
> |          | Accuracy                    | Response_length | Clarity | Accuracy             | Response_length | Clarity |
> | OTC-GRPO | 74.0                        | 1778            | 0.945   | 44.4                 | 1778            | 0.932   |
> | MO-GRPO  | 71.2                        | 1675            | 0.953   | 41.2                 | 1478            | 0.946   |
> | ParetoPO | 80.0                        | 1367            | 0.977   | 48.0                 | 1367            | 0.971   |
>
> [1] Lu, Y., Wang, Z., Li, S., Liu, X., Yu, C., Yin, Q., ... & Jiang, M. (2025). Learning to optimize multi-objective alignment through dynamic reward weighting.
>
> >**Q1: How sensitive is the initial phase of Stage 1 to the stochastic discovery of highly sub-optimal success paths, and does this risk trapping the policy in a local efficiency minimum?**
>
>  In early training, successful trajectories are sparse, so $N_{optimal}$ may be initialized from a successful but still inefficient trajectory. However, this does not trap the policy in a local efficiency minimum for three reasons:
>
> 1. First, as described in Section 2.3, $N_{optimal}$ is monotonically improved across epochs: whenever a successful trajectory with fewer tool calls is found, it is updated. Thus, it acts as a progressively tightened efficiency bound rather than a fixed bound.
> 2. Second, Stage 1 uses dynamic scalarization instead of a single reward, and its hypervolume-based weighting encourages exploration of under-covered regions in the objective space rather than staying near suboptimal solutions.
> 3. Third, Stage 2 further mitigates this issue through Pareto ranking. Once a trajectory is found that is equally correct but more tool-efficient, it will rank higher and receive a larger advantage.
>
> To support this explanation, we additionally provide average $N_{optimal}$ on NQ of Qwen2.5-3B (Base) over training epochs, together with the average tool-call count. We can see that $N_{optimal}$ quickly stabilizes and the optimization becomes approximately stationary in later stages.
>
> | training_step | 100 | 500 | 1000 | 1500 | 2000 |
> |---------------|-----|-----|------|------|------|
> | $N_{optimal}$     | 2.1 | 1.5 | 1.2  | 0.8  | 0.8  |
> | #tool_call    | 2.6 | 1.8 | 1.3  | 0.9  | 0.9  |
>
> >**Q2: Our approach achieves both higher accuracy and efficiency than baselines.**
>
> Our results suggest that many single-objective methods might be not on a strong Pareto frontier. In tool-using agents, excessive tool use can introduce noise, lengthen trajectories, and worsen credit assignment, so reducing redundant tool calls can improve both efficiency and accuracy.
>
> Theoretically, our method explicitly optimizes multiple objectives jointly: Stage 1 explores supported Pareto regions via dynamic scalarization, and Stage 2 uses Pareto ranking to favor nondominated trajectories and refine the policy toward Pareto-ascent stationary points. This helps ParetoPO move away from process-inefficient local optima toward a better Pareto region.
>
> >**Q3: More results on BrowseComp and Tau-Bench.**
>
> We add the results in BrowseComp by fine-tuning Qwen2.5-7B as below. We can see that our method can still outperform baselines.
>
> |          | BrowseComp |            |
> |----------|------------|------------|
> |          | Accuracy   | #tool_call |
> | OTC-GRPO | 10.5       | 14.1       |
> | MO-GRPO  | 9.6        | 16.3       |
> | ParetoPO | 12.4       | 12.2       |
>
> >**Q4: More results on other LLMs.**
>
> Please see the W2 response to **Reviewer i3Wo** for Llama3.1-8B-Instruct results.
>
> >**Q5: Efficiency analysis of the proposed optimization method.**
>
> Please see the W3 response to **Reviewer QmdP** for efficiency analysis.

---

> > ### Author Rebuttal · Reviewer_whi1 · 2026-04-05
> >
> > My concerns have been adequately addressed, and score is updated accordingly.

---

### Decision · Program_Chairs · 2026-04-30

**Decision:**

Accept (spotlight)

**Comment:**

This paper received positive evaluations from the reviewers, who agreed that:

* It addresses an important and practical problem of balancing multiple objectives (e.g., accuracy and tool efficiency) in tool-augmented LLM agents;
* The proposed ParetoPO framework is novel, well-motivated, and supported by solid theoretical analysis;
* The empirical results consistently demonstrate improved trade-offs between performance and efficiency.

Reviewers also noted a few areas for improvement:

* The evaluation is limited to two objectives, and it remains unclear how well the method scales to more complex multi-objective settings;
* Experiments are conducted on a limited set of models, leaving questions about generalization to larger or more diverse backbones.

During the rebuttal, the authors addressed concerns regarding clarity and experimental design, which increased confidence in the method’s effectiveness and contribution.

Overall, the paper makes a meaningful and well-supported contribution to multi-objective optimization for LLM agents. Therefore, I recommend accepting this paper for the conference.